EMBO
Molecular Medicine

# EGFL7 enhances surface expression of integrin α5β1 to promote angiogenesis in malignant brain tumors

Nevenka Dudvarski Stanković[1,2,3,†], Frank Bicker[1,2,3,†], Stefanie Keller[1,2,3,†], David TW Jones[3,4,5,6], Patrick N Harter[2,3,7], Arne Kienzle[1,8], Clarissa Gillmann[9], Philipp Arnold[10], Anna Golebiewska[11], Olivier Keunen[11], Alf Giese[12], Andreas von Deimling[3,4,13], Tobias Bäuerle[9], Simone P Niclou[11,14], Michel Mittelbronn[11,15,16,17], Weilan Ye[18], Stefan M Pfister[3,4,5,6] & Mirko HH Schmidt[1,2,3,*]

## Abstract

Glioblastoma (GBM) is a typically lethal type of brain tumor with a median survival of 15 months postdiagnosis. This negative prognosis prompted the exploration of alternative treatment options. In particular, the reliance of GBM on angiogenesis triggered the development of anti-VEGF (vascular endothelial growth factor) blocking antibodies such as bevacizumab. Although its application in human GBM only increased progression-free periods but did not improve overall survival, physicians and researchers still utilize this treatment option due to the lack of adequate alternatives. In an attempt to improve the efficacy of anti-VEGF treatment, we explored the role of the *egfl7* gene in malignant glioma. We found that the encoded extracellular matrix protein epidermal growth factor-like protein 7 (EGFL7) was secreted by glioma blood vessels but not glioma cells themselves, while no major role could be assigned to the parasitic miRNAs miR-126/126*. EGFL7 expression promoted glioma growth in experimental glioma models *in vivo* and stimulated tumor vascularization. Mechanistically, this was mediated by an upregulation of integrin α5β1 on the cellular surface of endothelial cells, which enhanced fibronectin-induced angiogenic sprouting. Glioma blood vessels that formed *in vivo* were more mature as determined by pericyte and smooth muscle cell coverage. Furthermore, these vessels were less leaky as measured by magnetic resonance imaging of extravasating contrast agent. EGFL7-inhibition using a specific blocking antibody reduced the vascularization of experimental gliomas and increased the life span of treated animals, in particular in combination with anti-VEGF and the chemotherapeutic agent temozolomide. Data allow for the conclusion that this combinatorial regimen may serve as a novel treatment option for GBM.

**Keywords** angiogenesis; EGFL7; endothelial cell; glioblastoma; integrin
**Subject Categories** Cancer; Neuroscience; Vascular Biology & Angiogenesis

## Introduction

Glioblastoma (GBM) is one of the most lethal tumors with a mean overall patient survival time of only 15 months after diagnosis due to inevitable recurrences (Fine, 2014). Current treatment options for GBM comprise surgical resection, radiotherapy, and chemotherapy (Seystahl *et al*, 2016). Glioblastomas are composed of a heterogeneous mixture of poorly differentiated neoplastic astrocytes with different genetic abnormalities. Their characteristic dependency on

1  Molecular Signal Transduction Laboratories, Institute for Microscopic Anatomy and Neurobiology, Focus Program Translational Neuroscience (FTN), Rhine Main Neuroscience Network (rmn²), University Medical Center of the Johannes Gutenberg University, Mainz, Germany
2  German Cancer Consortium (DKTK), Partner Site Frankfurt/Mainz, Germany
3  German Cancer Research Center (DKFZ), Heidelberg, Germany
4  German Cancer Consortium (DKTK), Partner Site Heidelberg, Germany
5  Hopp Children's Cancer Center at the NCT Heidelberg (KiTZ), Heidelberg, Germany
6  Department of Pediatric Oncology, Hematology & Immunology, Heidelberg University Hospital, Heidelberg, Germany
7  Neurological Institute (Edinger Institute), Goethe University, Frankfurt am Main, Germany
8  Laboratory of Adaptive and Regenerative Biology, Brigham & Women's Hospital, Harvard Medical School, Boston, MA, USA
9  Institute of Radiology, University Medical Center Erlangen, Erlangen, Germany
10  Anatomical Institute, Kiel University, Kiel, Germany
11  NORLUX Neuro-Oncology Laboratory, Department of Oncology, Luxembourg Institute of Health (L.I.H.), Luxembourg, Luxembourg
12  Department of Neurosurgery, University Medical Center of the Johannes Gutenberg University, Mainz, Germany
13  Department of Neuropathology, German Cancer Research Center (DKFZ), Heidelberg, Germany
14  KG Jebsen Brain Tumour Research Center, University of Bergen, Bergen, Norway
15  Luxembourg Centre for Systems Biomedicine (LCSB), University of Luxembourg, Esch-sur-Alzette, Luxembourg
16  Laboratoire National de Santé (LNS), Dudelange, Luxembourg
17  Luxembourg Centre of Neuropathology (LCNP), Dudelange, Luxembourg
18  Vascular Biology Program, Molecular Oncology Division, Genentech, San Francisco, CA, USA
   *Corresponding author. Tel: +49 6131 17 8071; Fax: +49 6131 1747 8071; E-mail: mirko.schmidt@unimedizin-mainz.de
   †These authors contributed equally to this work

angiogenesis led to several phase 2 trials of bevacizumab (Avastin®), a humanized antibody targeted against vascular endothelial growth factor (VEGF) in patients with recurrent GBM (Jain *et al*, 2007). However, results were disappointing because neither the AVAglio (Chinot *et al*, 2014) nor the RTOG 0825 trial (Gilbert *et al*, 2014) revealed an effect on overall patient survival. Nevertheless, bevacizumab increased the progression-free periods of GBM patients in these studies by 3–4 months and is therefore used as a treatment option in some countries, e.g., the United States and Switzerland (Hundsberger & Weller, 2017). These disillusioning results may be due to tumor adaptation and exploitation of alternative angiogenesis pathways upon bevacizumab treatment (Bergers & Hanahan, 2008; Keunen *et al*, 2011). Therefore, novel combinatorial treatment regimens could prove to be useful to improve anti-VEGF efficacy.

Previously, we demonstrated that the proangiogenic factor epidermal growth factor-like protein 7 (EGFL7) is associated with blood vessel formation in brain neoplasia (Nikolic *et al*, 2013). Physiologically, this protein is highly secreted by the developing vasculature but downregulated in the endothelium of adults (Nikolic *et al*, 2010). Elevated expression of EGFL7 may still be detected in angiogenic vessels during tissue repair or regeneration (Nikolic *et al*, 2010) as well as in several tumor types, including colon (Diaz *et al*, 2008), gastric (Luo *et al*, 2014), breast (Fan *et al*, 2013), kidney (Khella *et al*, 2015), liver (Wu *et al*, 2009), and brain tumors (Huang *et al*, 2010). Increased levels of EGFL7 have been linked to increased dissemination of several tumors (Luo *et al*, 2014; Hansen *et al*, 2015; Deng *et al*, 2016) and to a reduced median patient survival time in glioblastoma patients (Huang *et al*, 2010).

Current data suggest that EGFL7 promotes angiogenesis by the activation of integrin $\alpha_V\beta_3$ (Nikolic *et al*, 2013). However, it remained enigmatic how this interaction choreographs the complex sequence of integrins and transmembrane receptors that allows a cell to switch from extracellular matrix (ECM) adhesion to migration (Morgan *et al*, 2009). In particular, the movement of endothelial cells (ECs) on extracellular matrix proteins such as fibronectin (Fn) does not solely rely on $\alpha_V\beta_3$ but rather on a temporal and spatial changeover with integrin $\alpha_5\beta_1$ (Morgan *et al*, 2009). However, the molecular mechanism of this nexus has not fully been unraveled.

In addition to EGFL7, the microRNAs-126 and 126*, parasitic microRNAs encoded by intron 7 of the *egfl7* gene, have been described as major regulators of ECs and angiogenesis (Wang *et al*, 2008). As a matter of fact, these miRNAs have been suggested to be the sole angiogenic players within the *egfl7* gene (Kuhnert *et al*, 2008); however, this view has recently been challenged (Lacko *et al*, 2017). In glioma cells, miR-126/126* act as tumor suppressors by affecting the Akt signaling pathway (Luan *et al*, 2015). Contrary to EGFL7, low expression levels of miR-126 have been described to correlate with increased migration of glioma cells and worse clinical outcome (Feng *et al*, 2012; Xu *et al*, 2017). However, it is not known whether or not miR-126/126* act independently or are linked to the expression of their host gene *egfl7*.

In an attempt to determine the role of EGFL7 in GBMs, we found that (i) EGFL7 was majorly expressed in glioma blood vessels, (ii) promoted glioma angiogenesis and as a consequence growth, (iii) acted independently of miR-126/126*, and (iv) EGFL7-blockage, in particular in combination with VEGF inhibition and temozolomide (TMD), proved to be a useful tool for brain tumor treatment in experimental glioma models.

# Results

### Source and regulation of EGFL7 in glioma

Both EGFL7 and miR-126/126* (Fig 1A) have been linked to the clinical outcome of GBM patients as these molecules have been implicated in the regulation of several tumor types. Therefore, it was tested whether or not expression of EGFL7 and miR-126/126* occurred in glioma specimens. Initially, promoter methylation arrays were performed and promoter regions were assigned as previously described (Zhang *et al*, 2013). CpG sites in the EGFL7 and miR-126/126* promoters displayed divergent changes in the methylation pattern among GBM subgroups. While CpG island 1 within the EGFL7 promoter was found unmethylated in almost all glioma samples, CpG island 2 responsible for miR-126/126* expression was mostly methylated with the exception of the histone H3 G34-mutant GBM subgroup (G34R; Fig 1B). Data indicate that the *egfl7* promoter was usually accessible for subsequent EGFL7 transcription in glioma specimens while the miR-126/126* promoter was mostly not.

Subsequently, human EGFL7 mRNA levels in glioma specimens, human stemlike brain tumor-propagating cells (BTPCs) or glioma

**Figure 1. EGFL7 expression in glioma.**

A   The *egfl7* gene (about 13.8 kb) comprises 10 exons with seven of them encoding for the EGFL7 cDNA (lower left). Intron 7 harbors the parasitic miRNAs miR-126 and miR-126* (lower right).

B   Methylation arrays of primary glioblastoma (GBM) specimens revealed that CpG island 1 (*egfl7* promoter) was mostly unmethylated with the exception of some samples in the RTKII subgroup (*). CpG island 2 (*miR-126* promoter) was found methylated in most cases except the G34R subgroup (**). G34R and K27M: GBMs with mutations in a gene for the H3.3 histone variant; RTKI and II: GBMs with a mutation in the receptor tyrosine kinase type I (proneural subtype) and II (classical subtype); MES: GBMs with a deleterious mutation in the *NF1* gene (mesenchymal subtype); IDH1: GBMs with a mutation in isocitrate dehydrogenase 1.

C   Quantification of human EGFL7 by qRT–PCR in primary GBM biopsies and corresponding PDX xenografts upon implantation of GBM-derived organotypic spheroids (n = 3; Mann–Whitney *U*-test; mean ± SEM); B, GBM biopsy; G, generation.

D   Alignment of EGFL7 expression along with stromal (VWF, VEGFR1, VEGFR2, ACTA2, CD248, ITGAX, AIF1, CD68, CD4, and PTPRC) or tumor markers (EGFR, PDGFRA, VEGFA, VIM, and NES) in primary GBM biopsies or GBM-derived organotypic spheroids (PDX xenografts).

E   Immunohistochemical (IHC) analyses revealed EGFL7 expression in blood vessels (arrow) and neurons (arrowheads) of healthy gray matter. Expression in malignant brain tumor specimens was restricted to blood vessels, in particular with a large diameter (arrows). Scale bar represents 50 μm.

F   Quantitative scoring of EGFL7 staining in glioma specimens yielded about 25–40% EGFL7-positive blood vessels (n = 13 astrocytoma WHO° grade II, n = 25 astrocytoma WHO° grade III, n = 24 GBMs WHO° grade IV).

cells (GCs) were quantified by quantitative reverse transcriptase–polymerase chain reaction (qRT–PCR). Primary glioma specimens displayed significant levels of EGFL7, while BTPCs and GCs expressed little to no EGFL7 (Appendix Fig S1A). In parallel, all

samples underwent TaqMan analysis to quantify miR-126/126* levels; however, overall expression of both microRNAs was low and not significantly different among the groups (Appendix Fig S1B and C). Findings suggest that EGFL7 was indeed expressed in glioma

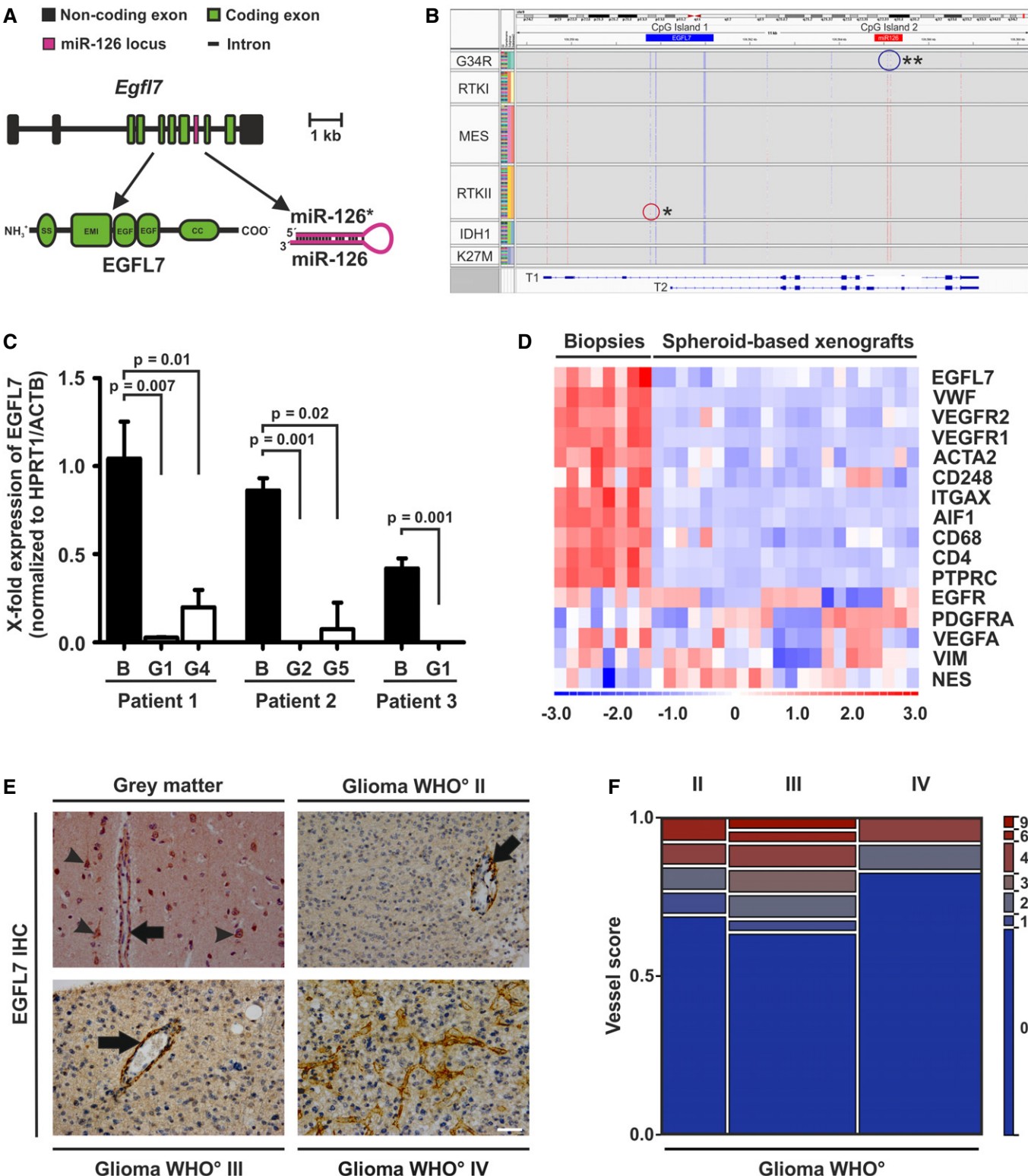

**Figure 1.**

specimens, while miR-126/126* were not, but expression was absent in glioma cells. In order to assess the methylation status of the *EGFL7* promoter in glioma cells, several BTPCs as well as LN229 cells were treated with a combination of DNA methyltransferase inhibitor 5-Aza-dC (Aza) and histone deacetylase inhibitor 4-phenyl-butyric acid (PBA). Upon treatment, EGFL7 was significantly upregulated in all cells as determined by qRT–PCR (Appendix Fig S1D). This indicated that the *egfl7* gene was indeed silenced, which raised the question whether or not EGFL7 expression is at all a characteristic of glioma cells.

Therefore, organotypic spheroids generated from patient-derived GBM biopsies (Golebiewska et al, 2013) were implanted into the brain of immune-deficient mice (Appendix Fig S1E). Human EGFL7 mRNA, as measured by qRT–PCR, was gone in all three xenografts already at early passages when compared to the primary human tumor (Fig 1C). This suggests that brain parenchymal or stromal but not glioma cells produced EGFL7. In order to test this, EGFL7, stroma and glioma marker expression was analyzed by qRT–PCR in patient-derived biopsies and the derived spheroid-based xenografts. Heatmap transformation followed by correlation analysis revealed that EGFL7 clustered with stromal markers (Fig 1D), e.g., vascular endothelial growth factor receptor 1 and 2 (VEGFR1/2) or von Willebrand factor (vWF) but not with tumor markers such as the epidermal growth factor receptor (EGFR) or vimentin (VIM). Data supported the fact that EGFL7 was indeed expressed in stromal cells but was absent from brain tumor cells.

Subsequent immunohistochemical analysis (IHC) of healthy brain and glioma specimens revealed that blood vessels (and neurons in the healthy brain) stained strongly positive for EGFL7 (Fig 1E). In particular, large blood vessels with a distinct lumen yielded a strong EGFL7 signal (Fig 1E, arrows). Glioma cells, however, did not stain positive above background. The specificity of the anti-human EGFL7 antibody applied was confirmed by costaining of glioma specimens with alternative anti-EGFL7 antibodies from different sources which yielded 100% overlap (Appendix Fig S1F). Quantification of EGFL7 staining in glioma specimens yielded 25–40% positive intratumoral blood vessels in each specimen (Fig 1F).

In conclusion, EGFL7 expression in glioma specimens was characteristic of blood vessels and occurred independently of miR-126/126*.

## EGFL7 in experimental glioma growth

To evaluate the role of EGFL7 in glioma formation *in vivo*, rodent GL261 glioma cells were lentivirally infected and as such manufactured to ectopically express human EGFL7 (hE7) or murine EGFL7 (mE7). Subsequent to fluorescence-activated cell sorting (FACS), immunoblotting of transduced GL261 cells confirmed ectopic EGFL7 expression in hE7 and mE7 GL261 cells as compared to negative control (Appendix Fig S2A). Subsequently, cells were intracranially implanted into the striatum of C57Bl/6 mice (Fig 2A) and all three groups were sacrificed after 24 days. Finally, the size of the resulting experimental gliomas was assessed by magnetic resonance imaging (MRI). Tumors ectopically expressing hE7 or mE7 were more than three times bigger than control tumors (Fig 2B). IHC using anti-hEGFL7 or anti-mEGFL7 antibodies confirmed retained ectopic expression of EGFL7 in hE7 and mE7 tumors *in vivo* (Appendix Fig S2B and C).

Alternatively, animals were sacrificed upon displaying first symptoms in a corresponding survival study. As expected, all animals developed tumors of similar sizes in this paradigm according to MRI (Fig 2C, left panel). However, mice bearing EGFL7-positive tumors died significantly earlier as compared to control (Fig 2C, right panel). The median survival time for mice bearing glioma with hE7 or mE7 expression was 29.5 days or 31.5 days, respectively, as compared to 34 days in the control group. Comparable results were obtained in a nude mouse paradigm bearing hE7- and mE7-expressing human U87 xenografts (Fig 2D; Appendix Fig S2D–F).

In order to evaluate the oncogenic potential of endogenous EGFL7 *in vivo*, GL261 glioma cells were intracranially implanted into the striatum of *EGFL7*-knockout (KO) mice (Schmidt et al, 2007) and respective wild-type (WT) litters. As this particular mouse model was found to manifest a deficit in miR-126/126* expression, the experiment was repeated in a mouse model lacking miR-126/126* only but retaining EGFL7 expression (Wang et al, 2008). All animals developed intracranial tumors, but the resulting median survival of *miR-126* KO mice and WT litters was not significantly different (Fig 2E). However, a general loss of EGFL7 significantly prolonged the median survival time of tumor-bearing mice from 35.5 days in WT littermates to 44 days in *EGFL7* KO animals (Fig 2F).

In order to assess the role of endogenous EGFL7 for glioma formation, BTPC11 cells underwent shRNA-based knockdown

---

**Figure 2. EGFL7 affected tumor growth and reduced overall survival.**

A   Schematic presentation of intracranial tumor implantation.
B   Overexpression of human EGFL7 (hE7) or murine EGFL7 (mE7) increased tumor growth (n = 9; one-way ANOVA).
C   Immunocompetent mice bearing GL261 tumors. Representative MRI images confirmed tumor implantation in all experimental groups (left). Kaplan–Meier curves revealed decreased survival of mice bearing EGFL7-expressing tumors (right; 34 days (control) vs. 29.5 days (hE7) vs. 31.5 days (mE7); n = 8; log-rank test).
D   Immunodeficient mice bearing U87 tumors. Representative magnetic resonance images confirmed tumor implantation in all experimental groups (left). Kaplan–Meier curves revealed decreased survival of mice bearing EGFL7-expressing tumors (right; 70 days (control) vs. 42.5 days (hE7) vs. 40 days (mE7); n = 8; log-rank test).
E   *miR-126* KO did not affect overall survival of GL261 tumor-bearing mice (n = 8).
F   *EGFL7* KO prolonged overall survival of GL261 tumor-bearing mice as compared to wild-type (WT) littermates (44 days in KO vs. 35.5 days in WT; n = 8 each; log-rank test).
G   Residual EGFL7 expression in BTPC11 glioma cells was reduced by a lentivirus-based approach. Knockdown of EGFL7 (shE7_1 or shE7_2) prolonged the median survival time of glioma-bearing mice (68 days or 69 days) as compared to scrambled control (shScr; 56 days; n = 6; log-rank test).
H   The blood vessel-specific and *miR-126*-independent KO of *EGFL7* in *EGFL7^{fl/fl};Cdh5-CreERT2* mice prolonged the overall survival of GL261 tumor-bearing mice as compared to WT littermates (38 days in KO vs. 33 days in WT; n = 7 for KO; n = 5 for WT; log-rank test).

Data information: All data are presented as mean ± SEM. Scale bars represent 5 mm.

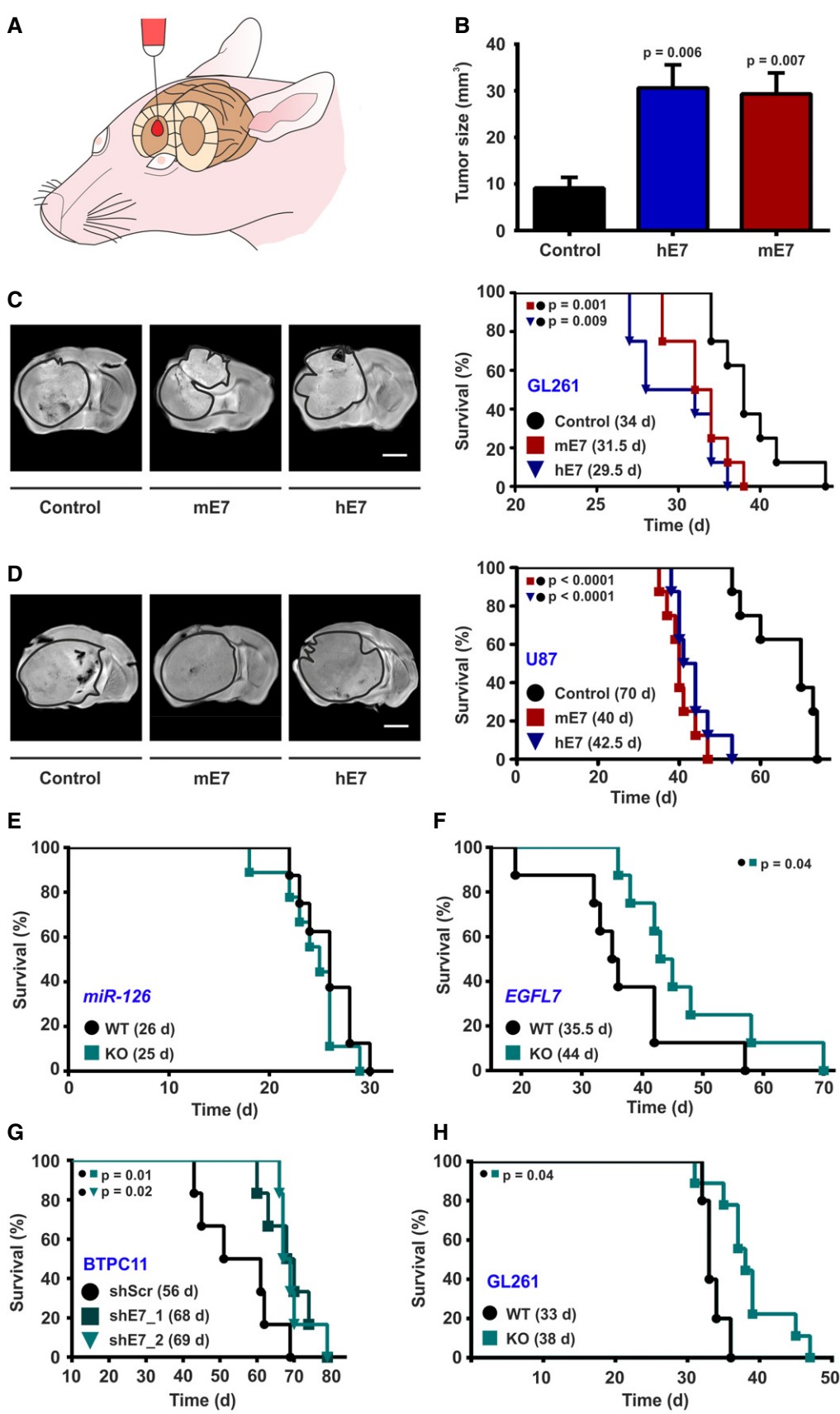

**Figure 2.**

studies to avoid affecting miR-126 expression. These cells were selected as they displayed quite low, yet detectable, levels of EGFL7 (Appendix Fig S1A). Two different lentiviruses encoding for human EGFL7-specific shRNAs (shE7_1 and shE7_2) were applied while a scrambled shRNA (shScr) virus served as a negative control. Western blot and qRT–PCR analyses verified that shE7_1 and shE7_2 silenced EGFL7 by > 90% or > 70%, respectively, as compared to control (Appendix Fig S2G and H). Subsequently, infected BTPC11 cells were intracranially implanted into immunodeficient mice and the median survival time was assessed. Knockdown of EGFL7 significantly prolonged the median survival time from 56 days in the shScr control group to 68 days in the shE7_1 and 69 days in the shE7_2 groups (Fig 2G).

Finally, the miR-126-independent oncogenic potential of EGFL7 was assessed by intracranial implantation of GL261 cells into the striatum of *EGFL7^fl/fl;Cdh5-CreERT2* mice. In this new model, application of tamoxifen allowed for the specific removal of EGFL7 from blood vessels but did not affect miR-126 expression (Bicker *et al*, 2017; Larochelle *et al*, 2018). The vessel-specific KO of EGFL7 significantly increased the median survival of tumor-bearing mice (38 days) as compared to 33 days in WT littermates (Fig 2H).

In conclusion, EGFL7 expression enhanced experimental glioma growth and decreased the life span of EGFL7-positive tumor-bearing mice.

## EGFL7-dependent growth and maturation of blood vessels in experimental glioma

Experimental glioma models were engaged to understand whether the influence of EGFL7 on EC was relevant for the vascularization of malignant brain tumors. U87 cells were engineered to ectopically express hE7 or mE7 by lentiviral infection as described above. After sorting, cells were intracranially xenografted into nude mice. Upon displaying first symptoms of sickness, animals were injected with the contrast agent Gadovist in the tail vein and sacrificed 15 min later. Brains were harvested and examined by IHC and MRI. Blood vessel IHC analysis and quantification by Imaris using the EC marker CD31 demonstrated significantly higher microvascular density in the presence of hE7 or mE7 (Fig 3A and B). T2-weighted MRI confirmed that all animals developed tumors (Fig 3C), and the analysis of T1-weighted images illustrated that in EGFL7-expressing tumors less Gadovist leaked into the glioma mass (Fig 3C and D). In the presence of an increased microvascular density, MRI findings suggested an increased maturation state of intratumoral blood vessels. In order to test this, maturation markers such as blood vessel coverage with pericytes (PDGFRβ; Fig 3E and F), smooth muscle cells (SMA; Fig 3G and H), or the presence of a basement membrane (Col IV; Fig 3I and J) were assessed by IHC. All

parameters were significantly increased in the presence of hE7 or mE7. Comparable results were obtained in the syngeneic GL261 glioma model (Appendix Fig S3A–J).

Data show that the presence of EGFL7 increased both the amount and maturation state of intratumoral glioma vessels.

## EGFL7 stimulates angiogenic sprouting in dependence of integrin α5β1

Previously, EGFL7 has been described as a proangiogenic factor (Lacko *et al*, 2017) that promotes angiogenic sprouting (Nikolic *et al*, 2013). In order to gain structural insights into this process, recombinant purified EGFL7 was analyzed using transmission electron microscopy (TEM). EGFL7 formed oligomers of varying sizes and displayed fibrillar structures (Fig 4A). High-magnification TEM images allowed the singling out of individual EGFL7 molecules and generation of class sums images. EGFL7 appeared as an elongated protein with two differentially charged poles (Fig 4A, lower right image). 30% of the primary structures resembled the light chain of coagulation factor VII and could therefore be molecularly modeled (Appendix Fig S4A). Among other features, the localization of the integrin-binding amino acid sequence RGD within the elongated part of the protein could be determined. Presumably, this motif is exposed once EGFL7 attaches to the ECM but is hidden in the soluble form of EGFL7. Furthermore, when extracting the charged amino acids from the EGFL7 sequence, EGFL7 was not homogeneously charged but contained a positive C- and a negative N-terminus (Fig 4B). This separation of charges enabled the formation of EGFL7 oligomers in the ECM in a head-to-tail fashion (Fig 4C).

Our previous observation that EGFL7 increased EC migration speed on Fn-coated surfaces depended on integrin αVβ3 (Nikolic *et al*, 2013) suggested an involvement of integrin α5β1 as it is required for rapid movement on Fn (Jacquemet *et al*, 2013). In order to determine whether EGFL7's proangiogenic effects are mediated by α5β1, recombinant purified EGFL7 was assessed *in vitro* in a 3D cell culture model of EC sprouting. The quantification of cumulative length of capillary-like structures revealed that EGFL7, Fn, and a combination of both significantly increased sprouting as compared to diluent (Fig 4D and E). However, this effect was blocked by the application of a specific α5β1-blocking antibody, indicating that EGFL7 indeed affected the EC-Fn interface via integrin α5β1.

To compare the influence of EGFL7 and Fn on integrin-dependent signaling in ECs, activation of classical downstream targets of integrin signaling, i.e., the Rho family GTPases Cdc42, Rac1, and RhoA, was analyzed in human umbilical vein endothelial cells (HUVECs) (Keely *et al*, 1997; White *et al*, 2007). EGFL7 alone activated Cdc42 (Fig 4F and G), but this effect was abrogated in the presence of Fn. In contrast, Fn preferentially activated Rac1, which

---

**Figure 3.  EGFL7 promoted density and maturation state of glioma vessels.**

A, B  CD31 staining for endothelial cells revealed increased tumor vascularization in mice bearing human EGFL7 (hE7)- or murine EGFL7 (mE7)-positive tumors (*n* = 3; one-way ANOVA).

C, D  T2-weighted magnetic resonance imaging (MRI) analysis of these brains confirmed that all mice developed tumors of similar size. T1-weighted MRI images showed decreased vessel permeability as measured by Gadovist extravasation in tumors expressing hE7 or mE7 (red arrows; *n* = 6; one-way ANOVA).

E–J  Enhanced maturation of glioma vessels in the presence of hE7 or mE7 was verified by increased co-localization of (E, F) PDGFRβ (pericytes), (G, H) SMA (smooth muscle cells), or (I, J) Col IV (basement membrane) with CD31 (*n* = 3; one-way ANOVA; quantifications normalized to CD31).

Data information: Data presented as mean ± SEM; AU, arbitrary units. Scale bars represent 60 μm (A, E, G and I) or 2.5 mm (C).

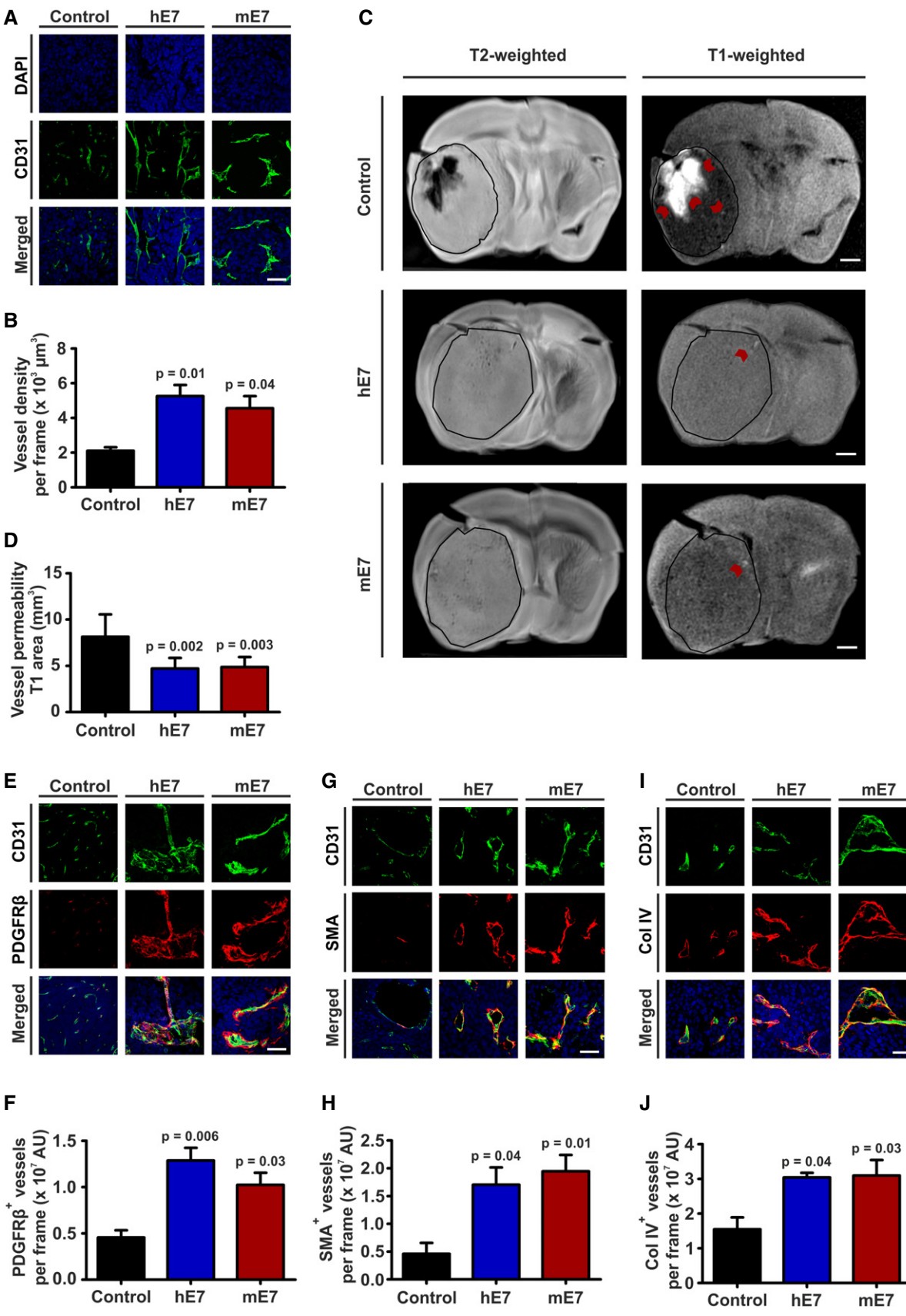

Figure 3.

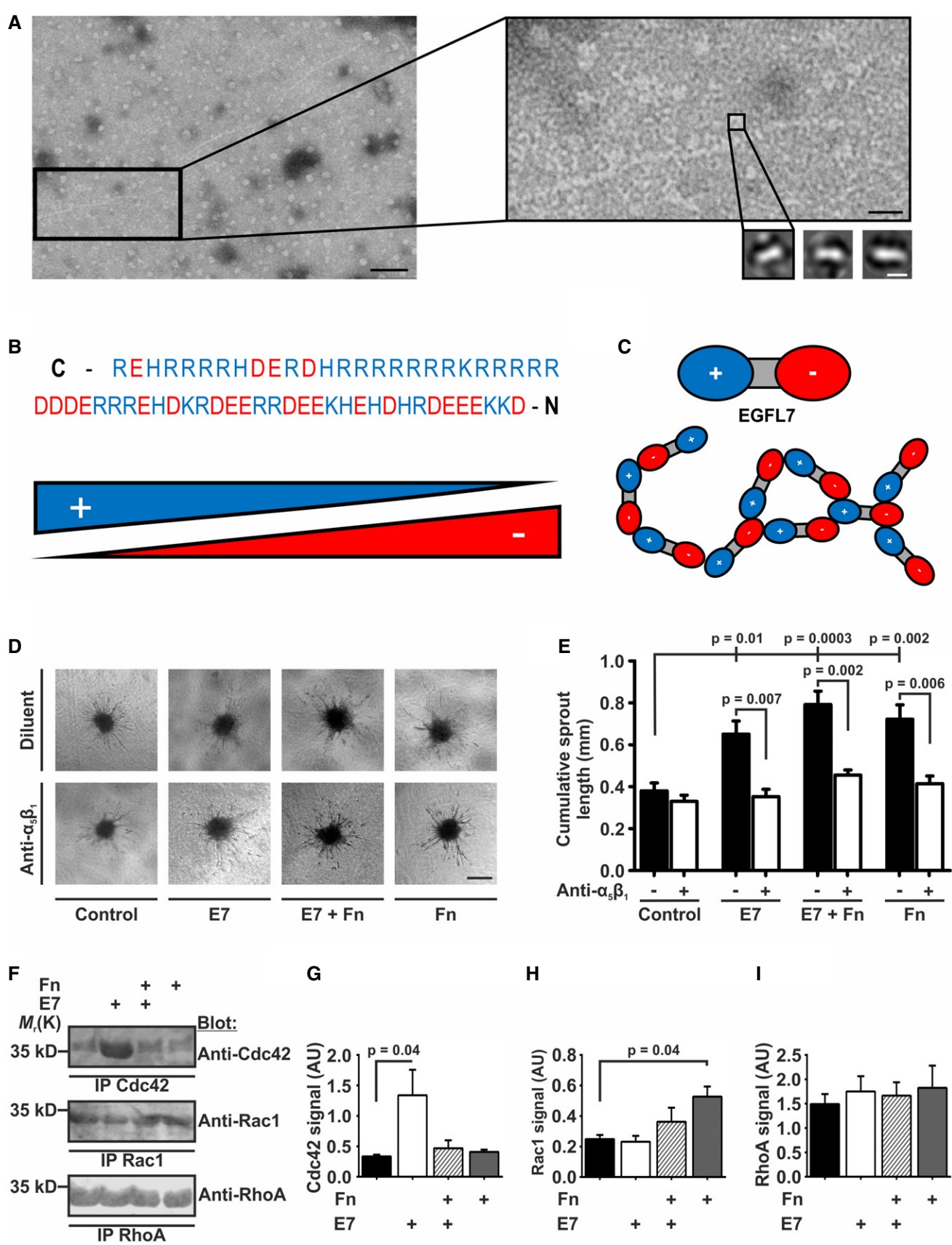

**Figure 4.**

◀

**Figure 4.  EGFL7 affected angiogenic sprouting in dependence of integrin $\alpha_5\beta_1$.**

A       Structural analysis of recombinant EGFL7 by transmission electron microscopy (TEM) showed aggregates with a fibrillar structure of varying sizes. Black scale bar left represents 50 nm. Higher magnifications depict rodlike EGFL7 particles. Black scale bar right represents 15 nm. White scale bar represents 5 nm.

B, C    Positively charged amino acid residues at the C-terminus and negative charges at the N-terminus make EGFL7 a polar molecule with a preference toward oligomerization.

D       Angiogenic sprouting of primary human umbilical vein endothelial cell (HUVEC) spheroids embedded in a collagen-based matrix was induced by the application of EGFL7 (E7), fibronectin (Fn), or a combination of both (upper row). Co-application of an integrin $\alpha_5\beta_1$-blocking antibody (anti-$\alpha_5\beta_1$) reduced sprouting in all cases but control (lower row). Scale bar represents 100 μm.

E       E7, Fn, and a combination of both significantly increased the mean cumulative sprout length per spheroid, an effect blocked by anti-$\alpha_5\beta_1$ ($n = 3$; one-way ANOVA (intergroup differences) and Mann–Whitney $U$-test (intra-group differences); mean ± SEM).

F–I     Measurement of Rho GTPase signaling downstream of integrins (F). Quantification of (G) Cdc42, (H) Rac1, and (I) RhoA activation by immunoblotting subsequent to seeding HUVECs on E7, Fn, or a combination of both. Preferentially, E7 activated Cdc42 and Fn activated Rac1, but both proteins antagonized each other. None of the treatments affected RhoA signaling ($n = 3$; one-way ANOVA; mean ± SEM). IP, immunoprecipitation; AU, arbitrary units.

Source data are available online for this figure.

was antagonized by EGFL7 (Fig 4F and H). Neither protein significantly affected RhoA activation (Fig 4F and I). Data suggest both EGFL7 and Fn affect HUVEC migration and adhesion via integrin receptors; however, they favor different Rho GTPases.

**EGFL7 increases surface expression of integrin $\alpha_5\beta_1$**

This raised the question how EGFL7 acted via integrin $\alpha_5\beta_1$ in the absence of a direct interaction (Nikolic *et al*, 2013). Recycling of both $\alpha_5\beta_1$ and $\alpha_V\beta_3$ has been shown to be linked (White *et al*, 2007; Caswell *et al*, 2008, 2009). Thus, we hypothesized that EGFL7 binds $\alpha_V\beta_3$, which in turn affects intracellular trafficking of $\alpha_5\beta_1$. To test this, the amount of both integrins located at the cellular surface was analyzed either by biotin labeling of surface-bound integrins or flow cytometry subsequent to labeling ECs with anti-$\alpha_V\beta_3$ or anti-$\alpha_5\beta_1$ antibodies (Fig 5A). Quantification of Western blots demonstrated increased levels of integrin subunit $\alpha_V$ in the plasma membrane after EGFL7 treatment alone or in combination with Fn, which itself did not alter the amount of surface-bound $\alpha_V$ or $\beta_3$, suggesting EGFL7 specificity (Fig 5B–D). Furthermore, treatment with EGFL7 or Fn significantly increased surface expression of subunits $\alpha_5$ and $\beta_1$. EGFL7 and Fn acted in an additive manner (Fig 5E–G). Results were confirmed using flow cytometry as EGFL7 treatment alone or in combination with Fn significantly increased the amount of surface-bound integrin $\alpha_V\beta_3$. Fn alone did not affect $\alpha_V\beta_3$ (Appendix Fig S4B and C); however, both EGFL7 and Fn increased surface-bound integrin $\alpha_5\beta_1$ to a comparable extent. Again, a combinational treatment with EGFL7 and Fn caused an additive effect (Appendix Fig S4D and E).

To test whether the increased levels of $\alpha_V\beta_3$ and $\alpha_5\beta_1$ on the cellular surface were due to an inhibition of endocytosis, colocalization of both integrins with early endosome antigen 1 (EEA1, marker of early endosomes) and lysosomal-associated membrane protein 1 (Lamp-1, marker of lysosomes) was investigated by IHC. Stainings were quantified by Imaris software. EGFL7 significantly diminished endosomal and lysosomal trafficking of integrin $\alpha_V\beta_3$, while Fn did not. However, it partially antagonized EGFL7's effects (Appendix Fig S5A–C). Intracellular trafficking of integrin $\alpha_5\beta_1$ was reduced by EGFL7, Fn, and combinations of both. However, Fn was slightly more relevant here as compared to the role of EGFL7 in $\alpha_V\beta_3$ trafficking. A combination of both ECM proteins reduced intracellular trafficking of integrin $\alpha_5\beta_1$ most effectively (Appendix Fig S5D–F).

Data suggest that EGFL7 and Fn interfered with the endocytosis of integrins $\alpha_V\beta_3$ and $\alpha_5\beta_1$, though to a different extent. EGFL7

displayed a preference for $\alpha_V\beta_3$ and Fn for $\alpha_5\beta_1$. In combination, both proteins increased the pool of surface-bound integrins and thereby allowed EC to migrate more rapidly on Fn in an $\alpha_5\beta_1$-dependent manner.

Moreover, GL261 glioma cells ectopically expressing mouse EGFL7 were intracranially implanted into the striatum of WT mice. Upon glioma implantation, these mice were treated twice a week by intraperitoneal injection of an $\alpha_5\beta_1$-inhibiting antibody or isotype control, which increased the survival of animals for about 4.5 days (Fig 5H), verifying that EGFL7 affected glioma growth dependent on integrin $\alpha_5\beta_1$.

**EGFL7-inhibition as an anti-angiogenic therapeutic for experimental glioma**

Data above offered the possibility that anti-EGFL7 treatment may affect the tumor vasculature of experimental glioma. Upon successful implantation and engraftment of tumors (U87 after 20 days, GL261 after 14 days), mice were intraperitoneally injected twice a week with antibodies targeting murine EGFL7 or VEGF, a combination of both or isotype controls as a negative control. Upon displaying symptoms of disease, animals were injected with Gadovist, sacrificed, and mouse brains assessed by IHC and MRI. U87 xenograft brain sections were stained for the endothelial marker CD31 to determine tumor vascularization, and for PDGFRβ, SMA, and Col IV to assess blood vessel maturation. CD31 was significantly lower in the groups treated with anti-EGFL7, anti-VEGF, or a combination of both blocking antibodies as compared to control, advocating treatment-driven inhibition of tumor vascularization (Fig 6A and B). Staining for PDGFRβ showed that blocking of EGFL7 or VEGF alone did not significantly influence the coverage of blood vessels with pericytes. In contrast, combinational treatment significantly decreased the amount of PDGFRβ-positive pericytes (Fig 6C and D). A significantly reduced amount of SMA-positive cells surrounding intratumoral blood vessels was spotted after treatment with anti-EGFL7 or anti-VEGF antibody. Combining both antibodies lead to the highest decrease in smooth muscle cell coverage (Fig 6E and F). Further, Col IV expression in the basement membrane of tumor blood vessels was significantly reduced after anti-EGFL7 and anti-VEGF antibody treatment. While anti-EGFL7 treatment exhibited stronger effects than anti-VEGF treatment, a combination of both antibodies displayed a significantly greater decrease in Col IV expression (Fig 6G and H). MRI analysis of the experimental brain tumors demonstrated an increased amount of Gadovist in the

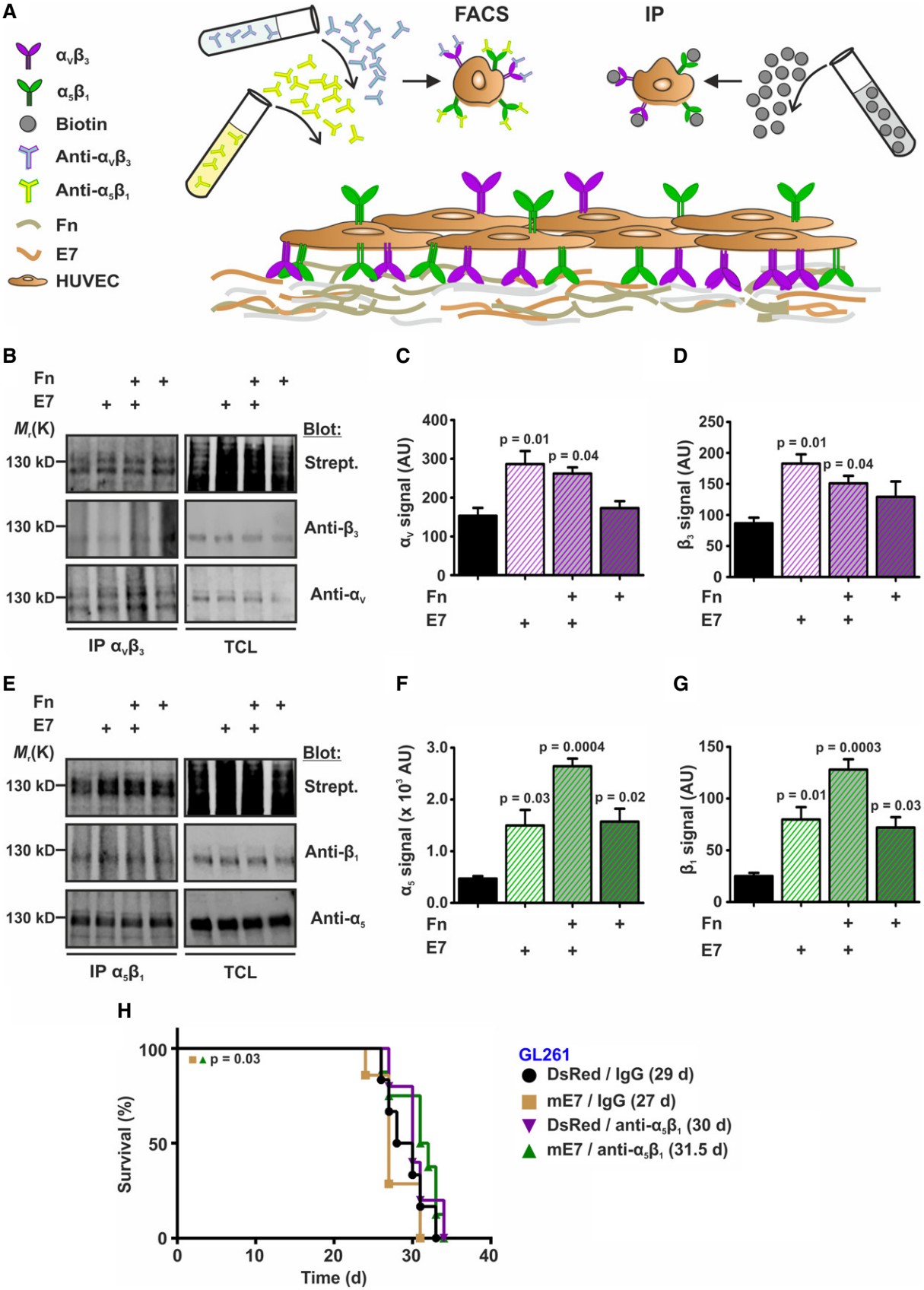

Figure 5.

**Figure 5.  EGFL7 increased surface expression of integrins $\alpha_5\beta_1$ and $\alpha_v\beta_3$.**

A   Schematic of the experimental setup to test for surface expression of integrins $\alpha_5\beta_1$ and $\alpha_v\beta_3$ in human umbilical vein endothelial cells (HUVEC). All surface molecules were marked by biotinylation followed by immunoprecipitation (IP) and Western blot specifically detecting each integrin $\alpha$ or $\beta$ subunit. Cells have not been permeabilized in either case to avoid staining of intracellular proteins. Alternatively, antibodies specific for both integrins were applied and detected by FACS.

B–D   Surface biotinylation assays verified increased surface expression of $\alpha_v$ and $\beta_3$ upon application of E7 but not Fn ($n$ = 3; one-way ANOVA).

E–G   Surface expression of $\alpha_5$ and $\beta_1$ was found increased upon treatment with E7 or Fn and was further enhanced by combination of both ($n$ = 3; one-way ANOVA).

H   Ectopic expression of murine EGFL7 (mE7) in GL261 mouse glioma cells followed by intracranial implantation into the striatum of immune-competent mice reduced overall survival but was rescued upon treatment with an integrin $\alpha_5\beta_1$-inhibiting antibody ($n$ = 6; log-rank test).

Data information: Data presented as mean ± SEM. TCL, total cell lysate; AU, arbitrary units.
Source data are available online for this figure.

glioma mass and brain parenchyma subsequent to anti-EGFL7, anti-VEGF, or combinational treatment, indicating increased blood vessel leakage upon inhibition of EGFL7 and VEGF (Fig 6I). Kaplan–Meier survival curves revealed that the inhibition of mouse EGFL7 significantly prolonged the median survival from 37 to 47 days or 51 days in case of anti-VEGF treatment. Simultaneous inhibition of EGFL7 and VEGF further increased the median survival time to 58 days (Fig 6J). Comparable results were obtained using the syngeneic GL261 model (Appendix Fig S6A–J).

In order to assess whether or not anti-EGFL7 treatment offers a benefit for glioma treatment in a clinical setting, the influence of a chemotherapy regimen on anti-VEGF and anti-EGFL7 treatment was studied subsequent to intrastriatal implantation of U87 in NOD SCID mice. Upon tumor engraftment, mice were treated with isotype control, anti-VEGF, or a combination of anti-VEGF plus anti-EGFL7 antibodies together with TMD. Mice receiving a combination of anti-VEGF and anti-EGFL7 antibodies survived significantly longer as compared to anti-VEGF- or isotype-treated control animals (Fig 6K).

In sum, inhibition of EGFL7 or VEGF alone or in combination resulted in fewer and less mature blood vessels in experimental gliomas as well as an increase in animal survival time, in particular in combination with TMD.

# Discussion

Glioblastoma is a highly malignant type of brain tumor with currently no curative treatment (Seystahl *et al*, 2016). Glioblastoma patients suffer from short survival, which requires novel treatment

options. This led to the development of bevacizumab, an antibody blocking VEGF-induced angiogenesis (Carmeliet & Jain, 2011) and thereby targeting the blood supply of malignant brain tumors (Jain *et al*, 2007). Unfortunately, this antibody failed in GBM trials and yielded no effect on overall survival (Fine, 2014). However, the increase in progression-free periods observed in these studies (Chinot *et al*, 2014; Gilbert *et al*, 2014) and the lack of treatment alternatives for malignant brain tumors drives physicians and researchers to continue using bevacizumab, making it a point of contention in the glioma field (Hundsberger & Weller, 2017).

In an attempt to improve the efficacy of VEGF inhibitors in blocking angiogenesis in malignant glioma, we explored the therapeutic potential of EGFL7 and miR-126/126* in this context. Each molecule has been previously studied in the context of angiogenesis (Soncin *et al*, 2003; Schmidt *et al*, 2007; Kuhnert *et al*, 2008; Wang *et al*, 2008; Nikolic *et al*, 2013; Bambino *et al*, 2014), and each has been suggested as a marker for the clinical outcome of GBM patients (Feng *et al*, 2012; Xu *et al*, 2017). However, a comprehensive functional study on the role of EGFL7 and miR-126/126* in glioma has been missing so far. Expression analysis revealed that EGFL7 was present in malignant glioma of various types but was lost in cultured glioma cells. Subsequent analyses revealed that EGFL7 secretion was indeed a feature of glioma blood vessels but not the tumor cells themselves. This finding is in disagreement with a previous study, showing significant EGFL7 expression in glioma cells of tumor specimens by IHC (Huang *et al*, 2010). However, the anti-EGFL7 antibody applied in this study had been optimized using U251 glioma cells as a positive control, a cell line that turned out to be negative for EGFL7 expression in our analyses. Putatively, the antibody used was not EGFL7-specific, which also puts the

**Figure 6.  Anti-EGFL7 treatment for the treatment of glioma *in vivo*.**

Glioma-bearing mice were injected with anti-EGFL7 or anti-VEGF antibodies, a combination of both (Combo), or isotype negative controls. Mice were sacrificed upon showing first symptoms of disease, and brains were analyzed by magnetic resonance imaging (MRI). Resulting brain tumor sections were analyzed for blood vessel density and maturation state by immunohistochemistry.

A, B   CD31 staining revealed that blocking of EGFL7, VEGF, or a combination of both leads to decreased tumor vessel density ($n$ = 3; one-way ANOVA).

C–H   Staining for (C, D) PDGFRβ ($n$ = 3; one-way ANOVA), (E, F) SMA ($n$ = 3; one-way ANOVA), or (G, H) Col IV ($n$ = 3; one-way ANOVA) revealed decreased amounts of blood vessel-associated pericytes, smooth muscle cells and Col IV in the basement membrane of blood vessels upon treatment with anti-EGFL7, anti-VEGF, and most significantly, a combination of both antibodies.

I   This resulted in an increased intratumoral vessel permeability as measured by T1-weighted MRI analyses of extravasating Gadovist ($n$ = 6; one-way ANOVA).

J   Treatment with anti-EGFL7 (47 days), anti-VEGF (51 days), or a combination of both antibodies increased the median survival time of glioma-bearing Rag1$^{-/-}$ mice (58 days) as compared to isotype-treated controls (37 days; $n$ = 6; log-rank test).

K   Treatment of glioma-bearing NOD SCID mice with temozolomide (TMD) as a chemotherapeutic agent and a combination of anti-EGFL7 and anti-VEGF antibody increased the median survival time (Combo; 87 days; $n$ = 5; log-rank test) as compared to anti-VEGF alone (80 days; $n$ = 5; log-rank test) or isotype-treated controls (69 days; $n$ = 5; log-rank test).

Data information: Data presented as mean ± SEM; AU, arbitrary units. Scale bars represent 60 μm.

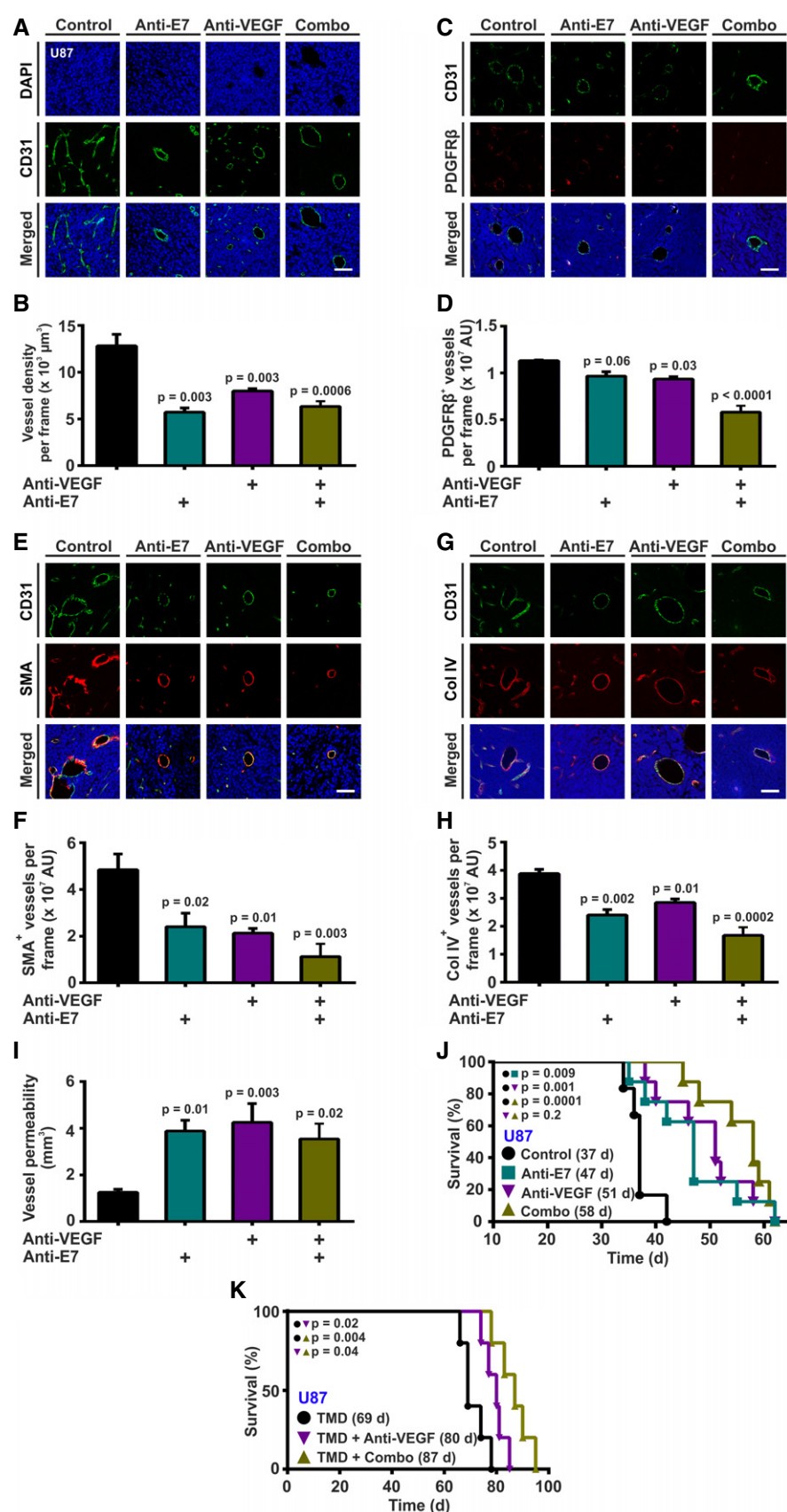

**Figure 6.**

conclusion of high EGFL7 levels being linked to reduced patient survival into jeopardy.

In parallel, we assessed the presence of miR-126/126* in glioma specimens. Though expression levels in primary tumors appeared slightly higher as compared to cultured cells, differences were not statistically significant. This agrees with previous observations describing low expression levels of miR-126 in glioma specimens (Han *et al*, 2016; Li *et al*, 2017; Xu *et al*, 2017). However, the fact that the *miR-126/126** promoter was mostly found methylated in glioma specimens and that there was no biological effect observed upon application of a syngeneic glioma model onto *miR-126/126** KO mice argues against a major role of these miRNAs in glioma. Nevertheless, an analysis of the G34R GBM subgroup for miR-126/126* might be promising as the *miR-126/126** promoter was found unmethylated in these samples. Furthermore, expression, promoter, and biological analyses allow for the conclusion that the expression of EGFL7 and miR-126 were not directly coupled in glioma specimens.

Increased expression of EGFL7 has been described for tumors of various origins (Diaz *et al*, 2008; Wu *et al*, 2009; Huang *et al*, 2010; Fan *et al*, 2013; Luo *et al*, 2014; Khella *et al*, 2015); therefore, we wondered about the influence of high EGFL7 levels on experimental glioma growth. The syngeneic GL261 as well as the U87 xenograft model have been selected due to their dependency on blood vessel growth, which makes them excellent tools to study the impact of EGFL7 on angiogenesis in a glioma-like environment. Both models displayed enhanced tumor growth rates as well as reduced survival upon the ectopic expression of EGFL7. Accordingly, increased tumorigenicity of EGFL7 was observed in models of breast and lung cancer (Delfortrie *et al*, 2011). *Vice versa*, implantation of GL261 cells into *EGFL7* KO mice increased animal survival arguing for a detrimental effect of high EGFL7 levels. Morphologically, tumors expressing high levels of EGFL7 displayed an increase in tumor vasculature and an increased maturation state of these vessels. This is in line with observations that describe EGFL7 as a regulator of vascular tube formation in ECs (Parker *et al*, 2004). However, convincing evidence for a role of EGFL7 in glioma angiogenesis *in vivo* was missing so far as previous reports on this topic restricted themselves to artificial 2D tube formation assays *in vitro*, which are not considered a reliable test for angiogenesis in the field (Huang *et al*, 2014a,b; Li *et al*, 2015). Furthermore, two of these studies were based on EGFL7 expression in U251 or U87 cells (Huang *et al*, 2014a,b), which both turned out to be EGFL7-negative in our analyses. The third study presented a strong immunohistochemical EGFL7 staining in neural stem cells (Li *et al*, 2015), which indeed are EGFL7-positive but produce only a tiny amount of this protein *in vitro* (Schmidt *et al*, 2009; Bicker *et al*, 2017). Furthermore, a recent study claims an EGFR-dependent effect of EGFL7 on glioma growth (Wang *et al*, 2017). Again, the EGFL7-negative cell lines U87 and U251 were analyzed in this study, this time for EGFL7 Western blots and IHC. Moreover, it has previously been shown that no direct interaction between EGFR and EGFL7 occurs upon overexpression and co-immunoprecipitation of both proteins (Schmidt *et al*, 2009). Although the conclusion of this last study showing that low EGFL7 levels improve survival of mice in an experimental glioma model is in agreement with our findings, it was compiled by EGFL7 knockdown in U87 cells. As mentioned above, these cells are EGFL7-negative; therefore, the conclusion of this study is not convincing. The use of inappropriate tools and techniques renders the abovementioned studies on the role of EGFL7 in GBM and glioma angiogenesis unreliable (Huang *et al*, 2010, 2014a,b; Li *et al*, 2015; Wang *et al*, 2017).

In order to understand the molecular mechanisms behind EGFL7's action, we focused on its role within the ECM, a key component of tumor angiogenesis (Weis & Cheresh, 2011). Structural analysis revealed that EGFL7 resides in the ECM as an oligomer with its extended structure exposing its central RGD integrin-binding domain. Previously, we have shown that this motif interacts with integrin $\alpha_V\beta_3$, resulting in enhanced migration speed on Fn (Nikolic *et al*, 2013). However, this is actually a feature of integrin $\alpha_5\beta_1$, while $\alpha_V\beta_3$ rather promotes slow and local migration (Jacquemet *et al*, 2013). This triggered the question how EGFL7 might affect integrin $\alpha_5\beta_1$ in the absence of a direct interaction (Nikolic *et al*, 2013). As the intracellular trafficking of both integrins had previously been connected (Caswell *et al*, 2008; Morgan *et al*, 2013), we determined the amount of surface integrin and found that the presence of EGFL7 along with Fn increased the amount of $\alpha_5\beta_1$ available at the cellular surface in an $\alpha_V\beta_3$-dependent manner. As described previously, this triggers focal adhesion maturation, hydrolysis of Rac1-GTP, and eventually, an increased migration speed of EC on an Fn surface (Morgan *et al*, 2009; Jacquemet *et al*, 2013), explaining the phenotype we observed before (Nikolic *et al*, 2013). This mechanism was verified by measuring Rho GTPase signaling downstream of integrins in HUVECs. While EGFL7 and Fn alone triggered strong activation of Cdc42 or Rac1, respectively, the combination of both annihilated this activation allowing for EC migration rather than adhesion. Interestingly, increased activation of Cdc42, as induced by EGFL7, has been reported to reduce the survival of brain tumor-bearing mice (Okura *et al*, 2016).

This influence of EGFL7 on integrin $\alpha_5\beta_1$ on the surface of ECs stimulated angiogenesis both *in vitro* and *in vivo*. EGFL7-expressing tumors inherited a denser and more mature vasculature covered with a greater amount of smooth muscle cells and pericytes, which as a result were less permeable. Reduced survival caused by the ectopic expression of EGFL7 in experimental gliomas was blocked by the specific inhibition of integrin $\alpha_5\beta_1$. Further, blocking EGFL7 with a specific antibody in experimental glioma models yielded a significant increase in median survival time. The resulting vasculature was more immature and more leaky. Combinational treatment with an anti-VEGF inhibitor, however, led to a significantly increased median survival time in both *in vivo* models, which is in accordance with results obtained in non-small-cell lung cancer models (Johnson *et al*, 2013). Whether or not this survival benefit in experimental glioma models will translate into an increased median survival of patients remains elusive. Analysis of publicly available data in the "R2: Genomics Analysis and Visualization Platform (http://r2.amc.nl)" revealed that human survival correlated positively with high EGFL7 levels in three databases, but negatively in two databases. Analyses of Rembrandt database data correlated high EGFL7 with shorter survival but at low case numbers. More detailed analyses including patient parameters such as age, GBM subtype, patient's pretreatment, or tumor grade will unravel this enigma. At this stage, it can be concluded that a combinatorial regimen of anti-EGFL7 and anti-VEGF antibodies increased the median survival in angiogenesis-dependent syngeneic and xenograft glioma models. However, beyond increasing the efficacy of bevacizumab, anti-EGFL7 treatment might serve to reduce bevacizumab-specific

toxic side effects such as hypertension or proteinuria. Usually, these manifest upon combination with chemotherapy in a dose-dependent manner (Afranie-Sakyi & Klement, 2015). A combination of both antibodies might be applied to reduce the doses of bevacizumab and/or chemotherapeutics to increase the well-being of patients suffering from glioma during their treatment, which is supported by our chemotherapy regimen combining TMD with anti-VEGF- and anti-EGFL7 treatment in experimental glioma.

# Materials and Methods

### Cell culture

Human umbilical vein endothelial cells were purchased from Lonza (Basel, Switzerland) and cultured as previously described (Nikolic et al, 2013). Human GCs G141, G142, G55, LN229, Mz18, U187, U251, U87, and mouse glioma cells GL261 were purchased from ATCC (Teddington, UK) and were cultured in Dulbecco's modified Eagle's medium (DMEM; Thermo Fisher Scientific, Waltham, MA, USA) supplemented with 10% fetal calf serum (Thermo) plus penicillin/streptomycin (Thermo). 1043, 1095, G112, GSC11, GSC2, GSC20, GSC23, GSC229, GSC240, and NSC7 II BTPCs were provided by Kenneth Aldape (Department of Pathology, University of Toronto, Canada) and Alf Giese (Department of Neurosurgery, University Medical Center Mainz, Germany). Cells were cultured at 37°C and 5% $CO_2$ in DMEM/F-12 medium (Thermo) supplemented with 1 mM HEPES (Sigma-Aldrich, St. Louis, Missouri, USA), 10 U/ml penicillin, 10 μg/ml streptomycin, 1× B27 supplement (Thermo), and 20 ng/ml EGF and bFGF (Peprotech, Rocky Hill, Connecticut, USA).

### TEM and molecular modeling of EGFL7

Recombinant EGFL7 was purified from Sf9 cells using a baculovirus system (Schmidt et al, 2009). Subsequently, TEM, molecular modeling and imaging TEM as well as class sum image calculations were performed (Arnold et al, 2014). Briefly, EGFL7 was added to a prior negatively glow discharged carbon-coated support grid (Science Service, Munich, Germany), stained with uranyl formate and transferred to a JEOL 1400Plus TEM machine operating at 120 kV. Images were taken, and individual EGFL7 molecules were singled out, aligned, and summed using IMAGIC 5 (Image Science, Berlin, Germany). Class sums represent 10–30 individual particles. Molecular modeling of the EGFL7 core domain was performed using the swiss-model workspace (Biasini et al, 2014). All molecular imaging was performed using UCSF Chimera (Pettersen et al, 2004).

### Surface biotinylation assay

Human umbilical vein endothelial cells were grown on dishes coated with 10 μg/ml EGFL7 and/or 4 μg/ml Fn. Cells were harvested after 2 h with accutase and were incubated in 0.93 mg/ml Sulfo-NHS biotin (Thermo) in phosphate-buffered saline for 30 min at room temperature. Likewise, HUVECs were treated for FACS-based integrin surface analyses but stained on ice for 45 min with 10 μg/ml mouse anti-$\alpha_V\beta_3$ (MAB1976, 1:100, Merck Millipore, Darmstadt, Germany) or 10 μg/ml mouse anti-$\alpha_5\beta_1$ (MAB1969, 1:100, Merck)

followed by Alexa Fluor 488-conjugated donkey anti-mouse antibody (1:600, Invitrogen). Dead cells were excluded by 4′,6-diamidino-2-phenylindole (DAPI) staining. All samples were analyzed using FACS Canto II (Becton Dickinson, Franklin Lakes, New Jersey, USA) and FlowJo v9.6.2 software (Tree Star, Ashland, Oregon, USA).

### Angiogenic sprouting

Angiogenic HUVEC sprouting was analyzed in vitro as previously described (Nikolic et al, 2013). Briefly, sprouting was induced by 10 μg/ml EGFL7, 4 μg/ml Fn (Sigma-Aldrich) or a combination of both and blocked by the co-application of 10 μg/ml anti-$\alpha_5\beta_1$-blocking antibody (MAB1969, Merck). Images were taken after 24 h on an inverted IX51 Olympus microscope (Hamburg, Germany), and cumulative sprout length was quantified using cellSens imaging software (Olympus, Hamburg, Germany).

### GTPase activation assay

Activation of small GTPases was analyzed using the Cdc42/Rac1/RhoA activation assay combo biochem kit (Cytoskeleton, Inc., Denver, CO, USA) according to the manufacturer's guidelines. In brief, HUVECs were grown for 2 h on wells coated with 10 μg/ml EGFL7, 4 μg/ml Fn or a combination of both. Subsequently, cells were harvested in ice-cold cell lysis buffer containing protease inhibitors. Lysates containing equal amounts of protein were incubated overnight at 4°C with rhotekin-RBD beads or PAK-PBD beads. Immunoprecipitates were analyzed by Western blot using mouse anti-Cdc42 (#ACD03, 1:250, Cytoskeleton), mouse anti-Rac1 (#ARC03, 1:500, Cytoskeleton), or mouse anti-RhoA (#ARH03, 1:500, Cytoskeleton).

### Epigenetic analyses

Epigenetic analyses, i.e., Aza and PBA cell treatment, were performed as described before (Saito et al, 2009). BTPC or LN229 cells were treated with 3 μM Aza (Sigma-Aldrich) for 24 h and 3 mM PBA (Sigma-Aldrich) for 5 days, and PBA was replaced on a daily base. Treatment was repeated for two consecutive cell passages. After each round of treatment, RNA was isolated from treated cells for qRT–PCR analysis using TRI reagent (Sigma-Aldrich) as previously described (Bicker et al, 2017). Primer sequences applied in SYBR Green-based (Bio-Rad, Hercules, CA, USA) analyses are summarized in Appendix Table S1. miR-126 and miR-126* were isolated by the TaqMan microRNA Reverse Transcription Kit (Thermo). Subsequently, qRT–PCR was performed using the KAPA PROBE FAST Universal qRT-PCR kit (Kapa Biosystems, Wilmington, MA, USA). miRNA expression was normalized to relative levels of endogenous RNU48.

### Immunoprecipitation and immunoblotting

The following antibodies were used for immunoprecipitation and Western blot studies (Bicker et al, 2017): mouse anti-tubulin alpha Ab-2 (MS-581-P1, 1:6,000, Thermo), goat anti-EGFL7 (R-12, sc-34416, 1:1,000, Santa Cruz Biotechnology, Dallas, TX, USA), rabbit anti-integrin $\beta_1$ (#4706, 1:1,000, Cell Signaling Technology, Danvers, MA, USA), rabbit anti-integrin $\beta_3$ (H-96, sc-14009, 1:1,000, Santa

Cruz), mouse anti-CD49e (610634, 1:1,000, BD Transduction Laboratories, San Jose, CA, USA), mouse anti-integrin $\alpha_V$ (P2W7, sc-9969, 1:1,000, Santa Cruz), IRDye 680RD-conjugated streptavidin, fluorescently labeled goat-anti-rabbit (1:15,000, LI-COR Biosciences, Lincoln, NV, USA), fluorescently labeled goat-anti-mouse (1:15,000, LI-COR Biosciences), and fluorescently labeled donkey-anti-goat (1:15,000, LI-COR).

### Immunofluorescence

Intracellular integrin trafficking was analyzed in HUVECs grown on 10 μg/ml EGFL7 and/or 4 μg/ml Fn-coated coverslips. Subsequently, cells were stained for EEA1, Lamp-1, and integrin $\alpha_5\beta_1$ or $\alpha_V\beta_3$. In brief, cells were fixed with 4% paraformaldehyde (PFA) and permeabilized with 0.1% Triton X-100. After blocking, cells were stained with sheep anti-EEA1 (AF8047, 1:100, R&D Systems, Minneapolis, MN, USA), rabbit anti-Lamp-1 (ab24170, 1:100, Abcam, Cambridge, MA, USA), mouse anti-$\alpha_5\beta_1$ (MAB1969, 1:100, Merck), or mouse anti-$\alpha_V\beta_3$ (MAB1976, 1:100, Merck) primary antibodies. Following incubation with Alexa Fluor 488-conjugated donkey anti-rabbit (1:1,000, Thermo), Alexa Fluor 647-conjugated donkey anti-sheep (1:1,000, Thermo), or Alexa Fluor 568-conjugated donkey anti-mouse (1:1,000, Thermo), nuclei were counterstained by DAPI staining. The coverslips were mounted using Fluoromount-G (SouthernBioTech, Birmingham, AL, USA), and images were taken using an SP8 confocal microscope (Leica, Mannheim, Germany). 3D reconstruction was performed using Imaris 8 (Bitplane, Zurich, Switzerland) and ImageJ software v1.41 (National Institute of Health, Bethesda, MD, USA).

### Lentiviral transduction

Lentiviruses were generated as previously described (Tiscornia et al, 2006). Successful gene transduction into GL261 or U87 cells was visualized by encoded tdTomato and into BTPC11 cells by Turbo-GFP. Lentiviruses encoding for human EGFL7-specific shRNAs (shE7_1 and shE7_2) or scrambled shRNA (shScr) were provided by Arie Reijerkerk (VU Medical Center, Amsterdam, Netherlands). Transduced cells were isolated in a single sort using a FACS Aria II (Becton Dickinson) device at the flow cytometry facility of the Institute for Molecular Biology, Mainz, Germany.

### Animal models

Animal experiments were approved by the ethics committee of the Landesuntersuchungsamt Rheinland-Pfalz, Germany, and conducted according to the German Animal Protection Law §8 Abs. 1 TierSchG. All mice were housed under specifically pathogen-free conditions at the Transgenic Animal Research Center (TARC, University Medical Center Mainz, Germany) on a 12-h dark/light cycle with unlimited access to food and water. All experiments were performed with 6- to 8-week-old male mice and were randomly allocated to the treatment groups. Rag1$^{-/-}$ and C57BL/6J-Tyrc-2J mice were provided by the TARC; NOD SCID mice were provided by the animal facility of the DKFZ.

Constitutive *EGFL7* KO mice, background C57BL/6J and 129/SvJ, were provided by Weilan Ye (Genentech, San Francisco, USA). In these animals, a retroviral gene trap vector containing stop codons in all three open reading frames was inserted upstream from intron 2 of the *egfl7* gene (Schmidt et al, 2007). However, miR-126 expression was found to be reduced by about 80% in this mouse line. Further, a novel EGFL7 KO model was created (*EGFL7$^{fl/fl}$;Cdh5-CreERT2*), background C57BL/6J, allowing for the specific deletion of EGFL7 in blood vessels upon the application of tamoxifen without affecting miR-126 expression (Bicker et al, 2017; Larochelle et al, 2018). Constitutive *miR-126* KO mice, background C57BL/6J and 129/SvJ, were provided by Marc Tjwa (Goethe University, School of Medicine, Frankfurt am Main, Germany). Here, the miR-126 locus in intron 7 of the *egfl7* gene was replaced with a neomycin resistance cassette flanked by loxP sites, which was later removed (Wang et al, 2008) allowing for the removal of miR-126 without affecting EGFL7.

### Intracranial implantations and treatment experiments

For stereotactic intracranial injections, immunodeficient Rag1$^{-/-}$ and NOD SCID or immunocompetent C57BL/6J-Tyrc-2J mice were anesthetized prior to intracranial implantation by intraperitoneal injection of 120 μl per 10 g body weight of a 12 mg/ml ketamine (Ratiopharm, Ulm, Germany) per 1.6 mg/ml xylazine (Bayer, Leverkusen, Germany) mixture. The anaesthetized animals were fixated in a stereotactic frame (Kopf Instruments, Tujunga, CA, USA), and a hole was drilled through the skull at the following coordinates relative to the bregma: 0.5 mm anterio-posterior and 2 mm medio-lateral. $2 \times 10^5$ U87 cells or $2.5 \times 10^3$ GL261 cells were injected 3 mm dorso-ventral of the dura mater into the striatum of C57BL/6J-Tyrc-2J, Rag$^{-/-}$ or NOD SCID mice, respectively. Mice were sacrificed by transcardial perfusion of a 4% PFA solution either after fixed time frames or upon the development of GBM-specific symptoms, such as lethargy, weight loss, and disheveled fur.

For treatment experiments, glioma-bearing animals received intraperitoneal injections of isotype control antibodies (10 mg per kg body weight (mpk) IgG1 and 5 mpk IgG2a), 10 mpk anti-EGFL7-blocking antibody, 5 mpk anti-VEGF-blocking antibody, or both anti-EGFL7-blocking and anti-VEGF-blocking antibodies twice a week after a time frame of 14 days (GL261 tumors) or 20 days (U87) of tumor engraftment. For combinatorial regimens, in addition to antibody treatment animals received 50 mg/kg TMD (Sigma, St. Louis, MO, USA) five times a week for 14 days by oral gavage (Hirst et al, 2013). For integrin $\alpha_5\beta_1$-inhibition experiments, C57BL/6J-Tyrc-2J mice were intrastriatally injected with GL261 glioma cells ectopically expressing murine EGFL7. Upon tumor engraftment, animals were treated twice a week with 7 mpk anti-mouse integrin $\alpha_5\beta_1$ or isotype control antibody. IgG1 and IgG2a isotype control antibodies, anti-EGFL7 (clone 18F7), anti-VEGF (clone B20-4.1.1), and anti-mouse integrin $\alpha_5\beta_1$ (clone 10E7) were generously provided by Weilan Ye (Genentech).

### IHC

Brains were removed and fixed in 4% PFA at 4°C overnight, subsequently transferred into a 30% sucrose solution and incubated at 4°C. Serial free-floating sections, 40 μm thick, were cut and incubated overnight with rat anti-CD31 (DIA-310, 1:100, Dianova, Hamburg, Germany), rabbit anti-collagen type IV (Col IV,

2150-1470, 1:300, Bio-Rad), mouse anti-SMA (A2547, 1:200, Sigma-Aldrich), rabbit anti-PDGFRβ (1:100, provided by William Stallcup, Sanford Burnham Prebys, La Jolla, CA, USA), goat anti-EGFL7 (AF3089, 1:100, R&D Systems), Armenian hamster-derived anti-EGFL7 (1:100, 1C8, provided by Weilan Ye, Genentech), primary antibodies followed by incubation with Alexa Fluor 488-conjugated goat-anti-rat (1:1,000, Thermo), Alexa Fluor 647-conjugated goat-anti-rabbit (1:1,000, Invitrogen), Alexa Fluor 647-conjugated goat-anti-mouse (1:1,000, Thermo), Alexa Fluor 647-conjugated donkey anti-goat (1:1,000, Thermo), or Cy3-conjugated goat-anti-Armenian hamster (1:1,000, Jackson ImmunoResearch Inc., West Grove, PA, USA) secondary antibodies. Cell nuclei were counterstained with DAPI staining. Images were captured using an SP8 confocal microscope (Leica). 3D reconstruction and analysis were performed using Imaris 8 (Bitplane) and ImageJ software v1.41 (National Institute of Health). Vessel density quantifications are represented in cubic microns ($\mu m^3$) obtained from the analysis of 3D image stacks. Vessel maturation was calculated as the fluorescence intensity sums of the angiogenesis markers PDGFRβ, SMA, or Col IV in close proximity to the blood vessel marker CD31 and plotted as arbitrary units.

## MRI studies

For MRI studies, 0.2 mmol/kg body weight MR contrast agent Gadovist (Bayer) was injected by intravenous injection in the tail vein 15 min before mice were sacrificed. Brains were removed, fixed in 4% PFA, and casted in 1.5% agarose to prevent movement within the scanner. Samples were placed in a mouse whole body coil (Bruker, Ettlingen, Germany) of a small animal ultra-high-field MR scanner (7 Tesla ClinScan 70/30). For morphologic images, the imaging protocol consisted of a 3D T2-weighted turbo spin-echo sequence (repetition time (TR) 2,000 ms, echo time (TE) 46 ms, averages (av) 2, acquisition time (TA) 6.5 h) and for contrast agent detection of a 3D fast low-angle shot T1 sequence (TR 50 ms, TE 2.5 ms, av 3, TA 1.8 h). On T2-weighted morphologic images, the tumor was segmented using Chimaera's segmentation tool (Chimaera GmbH, Erlangen, Germany) to determine the respective volumes. On T1-weighted images, hyperintense areas within the tumor were segmented to determine the volume of Gadovist leakage.

## Human glioma specimens

Human glioma biopsies were obtained from patients at the Goethe University Hospital Frankfurt, Germany. The use of tumor tissue was approved by the ethical committee of the Goethe University Hospital (GS04/09). Neuropathological diagnostics were performed by two experienced neuropathologists (PNH, MM) according to the classical WHO classification for tumors of the central nervous system. After histological examination of the human glioma biopsies, samples were frozen on dry ice and stored at −80°C until further processing.

10 μm cryosections of human glioma biopsies were prepared (Cryostat CM 1900, Leica, Wetzlar, Germany), dried for 30 min at 37°C, and incubated overnight with rabbit anti-EGFL7 (103-PA14, 1:100, ReliaTech, Wolfenbüttel, Germany) and goat anti-EGFL7 (AF3089, 1:100, R&D Systems) primary antibodies followed by

### The paper explained

#### Problem

Glioblastomas are typically lethal brain tumors causing patients' death within 1 year after initial diagnosis. Attempts have been made to increase survival time by the inhibition of tumor vascularization and thereby reduce oxygen and nutrient supply of tumors. Most unfortunately, these approaches exerted only a limited effect on patient survival, putting the applicability of angiogenesis inhibitors in glioma treatment *per se* into jeopardy.

#### Results

In order to overcome this shortcoming, we explored the role of a novel proangiogenic factor termed EGFL7 in malignant brain tumors. We detected the protein in tumor blood vessels and its forced expression promoted glioma growth by the enhanced formation of mature blood vessels in an integrin $\alpha_5\beta_1$-dependent manner. Inhibition of EGFL7 using specific antibodies reduced the vascularization of experimental brain tumors and increased survival.

#### Impact

We conclude from our study that a combinatorial regimen of anti-EGFL7 together with the angiogenesis inhibitor anti-VEGF and the chemotherapeutic agent temozolomide may serve as a novel treatment option for patients suffering from malignant glioma.

incubation with Alexa Fluor 488-conjugated donkey anti-rabbit (1:1,000, Thermo) and Alexa Fluor 568-conjugated donkey anti-goat (1:1,000, Thermo) secondary antibodies. Cell nuclei were counterstained with DAPI staining.

Staining intensity and frequency of EGFL7-positive human blood vessels were assessed using a previously established protocol (Harter *et al*, 2010). The frequency was determined with a semiquantitative score ranging from 0 to 4 (0 = 0–1%, 1 = 2–10%, 2 = 11–25%, 3 = 26–50%, 4 ≥ 50% of all cells showing positive staining). Staining intensity was recorded with a similar semiquantitative approach (0 = no signal, 1 = weak signal, 2 = moderate signal, 3 = strong signal). For the final score, the staining intensity and frequency scores were multiplied. Evaluation and photographic documentation was performed using an Olympus BX50 light microscope.

Patient-derived xenograft (PDX) samples were obtained by NORLUX (Luxembourg) and prepared as previously described (Bougnaud *et al*, 2016). Respective human glioblastoma samples were collected at the Centre Hospitalier in Luxembourg (Neurosurgical Department) and the Haukeland University Hospital (Bergen, Norway). Tissues were mechanically minced without enzymatic digestion, and organotypic spheroids were generated as previously described (Bougnaud *et al*, 2016). In brief, minced tumor samples were seeded on agar-coated dishes in DMEM (Lonza) with 10% fetal calf serum (Lonza), 2 mM L-glutamine (Lonza), 0.4 mM NEAA (Lonza), and 100 U/ml penicillin/streptomycin (Lonza). As soon as spheroids formed (after 7–10 days), six 300–400 μm large spheroids were implanted into the right frontal cortex of NOD SCID mice. Mice were sacrificed upon displaying first symptoms of sickness (locomotor problems, uncontrolled movements, prostration of hyperactivity). Xenografts were used for RNA isolation and re-implanted into several following generations. Written informed consent to use excessive tumor tissue for research purposes was obtained from

glioma patients. Use of the tissue was approved by the National Ethics Committee for Research Luxembourg and local ethics committee Haukeland University Hospital, Bergen.

## Statistics

Data were analyzed using GraphPad Prism 6 software (GraphPad Software, La Jolla, CA, USA). Unless otherwise indicated, all data are presented as mean ± 1 standard error of mean (SEM) of three independent biological experiments, animal experiments were performed with a minimum of six mice per group. Statistical analysis was performed using one-way analysis of variance (ANOVA) for three or more groups followed by *post hoc* analysis, or the Mann–Whitney *U*-test for comparison of two groups. For survival analyses, Kaplan–Meier survival curves were generated and compared using the log-rank test. Sample size and statistical methods used in the analyses are described in figure legends.

**Expanded View** for this article is available online.

## Acknowledgments

We thank Martin Adrian, Nikolai Schmarowski, Robert Varga, and Alexander Wenzel for their excellent technical assistance and Cheryl Ernest for proofreading the manuscript. This work was supported by the German Cancer Consortium (DKTK), Johannes Gutenberg University Medical Center (Stufe I to MHHS), the German Academic Exchange Service (DAAD) and Margareta Hugelschaffner Foundation (to NDS), the Luxembourg National Research Fond (FNR PEARL P16/BM/11192868 to MM) as well as the German Research Foundation (DFG) Collaborative Research Center 1292, project TP09 (to MHHS).

## Author contributions

NDS, FB, and SK designed and performed experiments; AGo, AK, CG, DTWJ, PA, and PNH performed experiments and analyzed the data; AGi, AD, MM, OK, SPN, SP, TB, and WY contributed expertise and tools; MHHS designed and supervised the study and edited the manuscript. All authors contributed to writing the manuscript.

## Conflict of interest

Weilan Ye is an employee and stockholder of Genentech/Roche. The remaining authors declare that they have no conflict of interest.

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
