## [Review Process File · EMBO Molecular Medicine]

EGFL7 enhances surface expression of integrin $\alpha 5\beta 1$ to promote angiogenesis in malignant brain tumors

Nevenka Dudvarski Stanković, Frank Bicker, Stefanie Keller, David T. W. Jones, Patrick N. Harter, Arne Kienzle, Clarissa Gillmann, Philipp Arnold, Anna Golebiewska, Olivier Keunen, Alf Giese, Andreas von Deimling, Tobias Bäuerle, Simone P. Niclou, Michel Mittelbronn, Weilan Ye, Stefan Pfister & Mirko H.H. Schmidt

Review timeline:

Submission date:	23 August 2017
Editorial Decision:	02 October 2017
Author's appeal:	10 October 2017
Editorial Decision:	30 October 2017
Revision received:	26 May 2018
Editorial Decision:	19 June 2018
Revision received:	02 July 2018
Accepted:	09 July 2018

Editor: Céline Carret

Transaction Report:

1st Editorial Decision

02 October 2017

Thank you for the submission of your manuscript "EGFL7 enhances surface expression of integrin $\alpha 5\beta 1$ to promote angiogenesis in malignant brain tumors". We have now heard back from the two referees whom we asked to evaluate your manuscript.

As you will see, while the referees acknowledge the potential interest of the study and reviewer #1 is rather on the optimistic side, referee #2 is much more reserved regarding the clinical and translational potentials of the strategy reported here. In fact this referee also remained on his/her position during our cross-commenting exercise (please see below), which, I am afraid, preclude publication of the manuscript in EMBO Molecular Medicine.

"I agree with the statement of Referee 1 "Inhibition of EGFL7 diminished tumor growth improving the survival of mice that were orthotopically harboring gliomas". However, I remain doubtful regarding the likely medical impact of the reported results based on the following considerations:

- 1) VEGF blockade is the current best option for anti-angiogenic treatment of glioma. Yet, it can only slightly prolong survival and cannot achieve a cure: therefore therapeutic targets that improve anti-VEGF potential are sorely needed.
- 2) In the data reported, anti-EGFL7 treatment performs similarly to anti-VEGF, but crucially the combination is not better than any of the two alone: the survival curves for the combined treatment are not statistically significantly better than single treatments in both Fig. 6J and Suppl. Fig. 5J. Further, all vascular parameters analyzed in both figures are affected essentially in the same way by both individual treatments and by the combination, in full agreement with the lack of advantage over anti-VEGF alone.

3) On the other hand, it is possible to observe a significant survival advantage over anti-VEGF alone in the same model with other treatments (e.g. data by the group of Rakesh Jain with an antibody targeting both VEGF and Ang2: see Fig. 1 in Klopper J. et al. PNAS 2016).

All in all, the data seem to suggest that EGFL7 and VEGF functions are not synergistic in glioma angiogenesis and so their individual targeting yields equivalent results, but does not show potential for improving the results of the gold-standard treatment with anti-VEGF alone even when combined.

Based on this, my evaluation is that the manuscript would be more suited for a journal with a more fundamentally oriented focus and less for EMBO Molecular Medicine, where the translational medical implications are center-stage."

I am sorry that the outcome for this manuscript could not have been more positive. I do want to emphasize, however, that this is not intended to imply a lack of interest on our part in either your work in particular or this field in general, and we hope that you will continue to consider EMBO Molecular Medicine for other submissions in the future when it seems appropriate.

***** Reviewer's comments *****

Referee #1 (Comments on Novelty/Model System for Author):

Currently anti-angiogenic approaches for the treatment of gliomas have confronted with major hurdles. The finding by the authors that EGFL7 might perform as a druggable pro-angiogenic factor is important and could have clinical application.

Referee #1 (Remarks for Author):

The authors show that EGFL7 by modulating the expression of avb3 and a5b1 on the tumor endothelial cells promotes glioma angiogenesis driving growth and invasion. Inhibition of EGFL7 diminished tumor growth improving the survival of mice that were orthotopically-harboring gliomas.

Comments:

The major strength of this paper is the use of orthotopic glioma models to assess the function of EGFL7. In addition, they have provided mechanistic data attempting to uncover the mechanism by which targeting various integrins might collaborate to block glioma cell growth.

There are a few minor issues that need to be addressed before publication of this paper.

- 1) The combination of anti-VEGF-A along with targeting EGFL7 might significantly increase the toxicity of this regimen. Could the authors discuss and propose an approach that will diminish the toxicity associated with this combination therapy?
- 2) EGFL7 could interfere with Notch signaling through blocking the activities of various Notch ligands, such as Jagged1 and Dll4. Then targeting EGFL7 might unleash Notch signaling, such as Dll4 activation of Notch1 and Notch 4 that could lead to non-productive angiogenesis. Thus, does targeting EGFL7 affect Notch dependent endothelial sprouting?
- 3) The precise mechanism by which EGFL7-mediated upregulation of avb3 and a5b1 regulate glioma tumor growth should be better defined.

Referee #2 (Remarks for Author):

The Authors explored the role of the extracellular matrix protein EGFL7 in driving angiogenesis in malignant glioma, a type of tumor that is highly dependent on vascular invasion and is typically lethal, and its therapeutic potential to improve the unsatisfactory results of anti-angiogenic therapy

with VEGF blockade. They found that EGFL7 is expressed in the blood vessels of human glioma samples, but not by the glioma cells themselves. Implantation of glioma cell lines in mice lacking EGFL7 prolonged survival of mice, whereas the absence of the EGFL7-encoded parasitic miRNA 126/126* did not have an effect. Conversely, EGFL7 overexpression in the glioma lines shortened survival, causing increased vessel density and also vessel maturation. Further, EGFL7 stimulated in vitro endothelial sprouting by increasing surface expression of integrin $\alpha 5\beta 1$ and reducing its endocytosis, possibly through interaction with integrin $\alpha v\beta 3$. Lastly, treatment with an EGFL7 blocking antibody reduced both vessel density and maturation in experimental gliomas and prolonged the median survival time of implanted animals. This effect was similar to that of a VEGF blocking antibody and the combination, although modestly increasing median survival time, did not show significant synergistic effects.

This study addresses a significant problem, namely the lack of effective treatments for malignant glioblastoma, and reports some interesting results. However, a few considerations limit the general importance of the findings:

- 1) The medical impact appears modest. In fact, while the data show that EGFL7 does have a role in glioma angiogenesis, its targeting is not more effective than the established (albeit still unsatisfactory) approach of VEGF blockade and their combination does not significantly improve the outcome of anti-VEGF alone. Further, the combination treatment does not significantly improve any vascular parameter compared to anti-VEGF or anti-EGFL7 alone, including vessel density, suggesting there is no additive or synergistic anti-angiogenic effect, in agreement with the modest and not-significant effect on duration of survival.
- 2) The mechanistic part, showing the effects of EGFL7 on integrin recycling and in vitro endothelial sprouting, is limited in scope. On one hand, the data provide only an incremental step in knowledge compared to previous results by the same group, and on the other they fail to address the actual relevance of these findings for glioma therapy. Does targeting of $\alpha 5\beta 1$ recapitulate the effects of anti-EGFL7 treatment in experimental glioma models? Or is the effect of EGFL7 targeting in gliomas due to other EGFL7 functions that have been described, for example its regulation of Notch signaling?
- 3) The proposed molecular mechanism that EGFL7 increases surface expression of $\alpha 5\beta 1$ integrin through ligation with $\alpha v\beta 3$ makes sense, but it would need to be proven by a loss-of-function experiment, showing that interference with $\alpha v\beta 3$ by a blocking antibody or siRNA knockdown does prevent the effects of EGFL7 on $\alpha 5\beta 1$.
- 4) The data show that EGFL7 targeting reduces vessel maturation and increases permeability, but no attempt is made to address how EGFL7 may regulate these processes.

Other points

- 1) Fig. 1B - It is unclear how to interpret this panel: why do red circles appear to indicate methylation in CpG island 1 and no methylation in CpG island 2?
- 2) Suppl. Fig. 1D - It would be useful to include a scale also in the lower part of the y axis to interpret the range of the black bars: do they reach to 1? 2? 5?
- 3) Suppl. Fig. 1E - I could not find any method for the derivation of organotypic spheroids from patient-derived GMB biopsies.
- 4) Suppl. Fig. 1F - The EGFL7 stains are difficult to interpret, possibly due to low magnification, and also appear to delineate dot-like structures rather than vessels, like evident instead in the histochemistries shown in Fig. 1E. Higher-magnification and clearer images would be helpful.
- 5) The conclusion that EGFL7 expression in glioma specimens was restricted to blood vessels is substantiated by the data, but that it occurred independently of miR-126/126* would be strengthened by similar analysis of its pattern of expression (or lack thereof) in the patient samples by in situ hybridization.

6) Suppl. Fig. 2B, C and F - The pattern of EGFL7 appears clearly suggestive of vascular structures, although the glioma cells are responsible for its overexpression in these over-expressing tumors. Curiously, instead, the pattern in panel 2E is not vascular anymore. This localization should be verified by co-staining with endothelial and glioma markers and an explanation would be useful.

7) Fig. 5D-E - The provided MRI images appear to serve just a decorative function, since only one condition is shown and it is not actually mentioned which one it is. On the other hand, the graph in panel A is rather superfluous (intrastratial injection and subsequent histological analyses are straightforward and were already described in a cartoon in Fig. 2A) and it could be removed, using the space to show complete MRI data for all conditions.

8) Figs. 2 and 5 analyze the same experiment and would best follow each other rather than being separated by the integrin data.

9) How were histological quantifications of vascular parameters actually performed? Vessel density is presented as cubic microns, which seems more a unit of volume than of density. What are the arbitrary units used to score vascular maturation? This information is missing from the methods and quite important to understand what is being measured and its significance. In fact, for example, just looking at the immunofluorescence images in Fig. 5 and Suppl. Fig. 4, pretty much all endothelial structures appear associated with collagen IV, SMA and PDGF-Rb signals in all conditions (except the single vessel shown in the control of Fig. 5G). This would suggest that there are no "naked vessels", but then quantifications in Arbitrary Units show dramatic differences for all these parameters.

Author's appeal

10 October 2017

Thank you for your consideration of our manuscript for EMBO Mol Med. I was pleased reading how much merit reviewer 1 found in our work. However, I was fairly surprised on the judgement of reviewer 2, which you eventually followed.

The reviewer's claim that any new anti-angiogenesis treatment of glioma can only be an additive to anti-VEGF treatment as the gold standard does not hold true for glioma patients. As we discuss in our manuscript the AVAglia and RTOG 0825 studies very convincingly show that anti-VEGF treatment remains palliative. The patients have no overall survival benefit but if at all their symptoms are slightly reduced within a narrow time window. Therefore bevacizumab (Avastin®), the best-known VEGF-inhibitor, has not been accredited for glioma treatment by the EMA in the EU. Only the FDA approved it in the US and it is used in Switzerland as a palliative treatment. In light of these studies it seems unlikely that anti-VEGF-treatment will remain a significant option for the treatment of malignant glioma in the EU. Certainly, it is far away from being a gold standard though it is still used in the absence of better alternatives, which are desperately needed as patients usually within one-year post diagnosis.

Anti-EGFL7-treatment seems quite promising in this context as it increased the survival time in our glioma models comparably to anti-VEGF treatment. Though reviewer 2 is right that the add on effect reached by a combination of both treatments (Fig. 6J and Suppl. Fig. 5J) was not highly significant ($P=0.16$ for the GL261 model and $P=0.29$ for U87), a strong tendency towards improvement of the anti-VEGF regimen by anti-EGFL7 could be observed. However, the potential improvement of anti-VEGF efficacy was just a side observation and we haven't followed up on it deeper as it was our goal to identify EGFL7 as a target for glioma treatment per se. If you would grant us the opportunity to increase the sample size of this particular experiment I am convinced that this difference would become statistically more significant as it is now.

In addition, the EGFL7 protein inherits some promising advantages over VEGF as a target molecule. EGFL7 is a secreted and barely soluble protein (comparable to fibronectin) which resides within the extracellular matrix of the tumor once the blood vessels are gone upon therapy. There it supports the regrowth of blood vessels along the trails of the former blood vessels during tumor recurrence. Blocking it delays blood vessel re-growth. In particular, this is relevant for malignant glioma as in this type of neoplasm the recurrence kills the patient, not the primary tumor. Therefore,

anti-EGFL7 treatment does not only act comparably to anti-VEGF treatment (and likely improves it) but in addition offers some remarkable advantages over it.

In light of these explanations I respectfully ask you to revisit your decision on our manuscript in order to give us the opportunity to present anti-EGFL7-treatment as a promising new option for the cure of malignant glioma. We believe that our findings are important for patients and will significantly contribute to their cure.

2nd Editorial Decision

30 october 2017

Thank you for your patience while we were reconsidering our decision. I am sorry it took so long, we are short in staff at the moment, hence the delay.

I have asked one of our external advisors about your paper and I am happy to disclose that this advisor agrees with you and referee 1 that the paper should be revised. Therefore, I'd like to invite you to revise the paper but please pay attention to referee 2's comments and address the commented limitations as much as possible. I would also like for you to try and address our advisor suggestions as we agree it would improve the clinical relevance of the findings.

"... the lack of a synergistic effect could be analyzed by studying the effect of EGFL blocking on VEGF production. Because anti-angiogenic approaches are not used as single agents in these tumors, it should be more interesting to compare the following arms: 1) standard chemo; 2) anti-VEGF; 3) anti- EGFL; 4) 2) anti-VEGF- chemo; 3) anti EGFL+chemo"

We would welcome the submission of a revised version within three months for further consideration and would like to encourage you to address all the criticisms raised as suggested to improve conclusiveness and clarity. Please note that EMBO Molecular Medicine strongly supports a single round of revision and that, as acceptance or rejection of the manuscript will depend on another round of review, your responses should be as complete as possible.

I look forward to receiving your revised manuscript.

2nd Revision - authors' response

26 May 2018

Editor (Remarks for Author):

Because anti-angiogenic approaches are not used as single agents in these tumors, it should be more interesting to compare the following arms: 1) standard chemo; 2) anti-VEGF chemo; 3) anti-VEGF anti EGFL chemo

Response to the editor

In order to address the interesting suggestion of the editor, we established a regimen using temozolomide as a chemotherapeutic agent in combination with anti-VEGF and anti-EGFL7 antibodies. Upon tumor engraftment, animals were treated twice a week with this combination therapy until the end of the experiment. Mice receiving anti-VEGF and anti-EGFL7 antibodies in combination with temozolomide survived significantly longer as compared to anti-VEGF (survival increased by about 7 d on average) or isotype control (survival increased by about 18 d on average)

treatment alone (new Fig. 6K). Data suggest a beneficial effect of EGFL7 treatment on standard glioma therapy and underpin the clinical relevance of our findings.

Referee #1 (Remarks for Author):

1) The combination of anti-VEGF-A along with targeting EGFL7 might significantly increase the toxicity of this regimen. Could the authors discuss and propose an approach that will diminish the toxicity associated with this combination therapy?

Response to the reviewer

The reviewer's point is well taken. Bevacizumab-specific toxicities such as hypertension or proteinuria manifest upon combination of the antibody with chemotherapy. The incidence of hypertension for example increases in a dose-dependent manner. Anti-EGFL7 treatment could serve to reduce the anti-VEGF concentration and thereby reduce the specific toxic side effects of bevacizumab. Furthermore, a combination of anti-VEGF/anti-EGFL7 might be applied to reduce the doses of chemotherapeutics. This point has been added to the discussion in the revised version of the manuscript.

2) EGFL7 could interfere with Notch signalling through blocking the activities of various Notch ligands, such as Jagged1 and Dll4. Then targeting EGFL7 might unleash Notch signalling, such as Dll4 activation of Notch1 and Notch 4 that could lead to non-productive angiogenesis. Thus, does targeting EGFL7 affect Notch dependent endothelial sprouting?

Response to the reviewer

The reviewer is completely right; EGFL7 affects angiogenesis in a Notch-dependent manner. This has been shown by Heidi Stuhlmann's group at Cornell University (Nichol et al., Blood, 2010). However, whether or not signalling mechanisms in physiological and pathological angiogenesis may be compared is still under debate. In particular, the role of Notch signalling in glioma per se has not been unravelled, yet. Others (Koch & Radke, Cell Mol Life Sci, 2007) and us (Teodorczyk & Schmidt, Front Oncol, 2015) elaborated on this topic in reviews as Notch may act as an oncogene but also as a tumour suppressor. In order to interpret our findings on EGFL7 in glioma in a Notch-dependent context it would be necessary to first understand not only the role of Notch receptors and ligands but also Notch signalling in general in malignant brain tumours. This is certainly a fascinating topic, but too complex to be addressed in the time frame of this revision and likely difficult to answer conclusively as questions concerning Notch in glioma have persisted for some time.

3) The precise mechanism by which EGFL7-mediated upregulation of $\alpha V\beta 3$ and $\alpha 5\beta 1$ regulate glioma tumour growth should be better defined.

Response to the reviewer

In order to address the reviewer's concern we analyzed Rho GTPase signaling downstream of integrins. In primary endothelial cells, EGFL7 and Fn preferentially activated Cdc42 or Rac1, respectively. In combination, both proteins annihilated each other, allowing for cell migration rather than adhesion and thereby supporting our hypothesis on EGFL7's influence on coordinated $\alpha V\beta 3/\alpha 5\beta 1$ trafficking (Fig. 4F-I). Furthermore, GL261 glioma cells ectopically expressing EGFL7 were implanted in the striatum of C57BL/6 mice, which were treated with an $\alpha 5\beta 1$ integrin-inhibiting antibody twice a week upon tumour implantation. Treatment significantly increased the survival time of the animals by 4.5 days on average (Fig. 5H).

Referee #2 (Remarks for Author):

1) The medical impact appears modest. In fact, while the data show that EGFL7 does have a role in glioma angiogenesis, its targeting is not more effective than the established (albeit still unsatisfactory) approach of VEGF blockade and their combination does not significantly improve the outcome of anti-VEGF alone. Further, the combination treatment does not significantly improve any vascular parameter compared to anti-VEGF or anti-EGFL7 alone, including vessel density, suggesting there is no additive or synergistic anti-angiogenic effect, in agreement with the modest and not-significant effect on duration of survival.

Response to the reviewer

Though we appreciate the reviewer's opinion, we are convinced that our findings have major medical relevance and that anti-EGFL7 treatment could be an alternative to anti-VEGF treatment. The reviewer's comment implies that anti-EGFL7 might be an add-on to anti-VEGF treatment only. However, as we discuss in our manuscript, the AVAglia and RTOG 0825 studies very convincingly show that anti-VEGF treatment remains palliative in glioma patients. The patients have no overall survival benefit, and at best their symptoms are slightly reduced within a narrow time window. Therefore, bevacizumab (Avastin®), the best-known VEGF-inhibitor, has not been accredited for glioma treatment by the EMA in the EU. Only the FDA approved it in the US and it is used in Switzerland as a palliative treatment. In light of these studies, it seems unlikely that anti-VEGF-treatment will remain a significant option for the treatment of malignant glioma in the EU. Other anti-angiogenesis treatments seem the logical alternative.

Instead, anti-EGFL7 treatment seems promising and increased the survival time in our glioma models comparably to anti-VEGF (Fig. 6 and Suppl. Fig. 6). Although Reviewer 2 is right that the add-on effect on animal survival reached by a combination of both treatments was not highly significant ($P = 0.17$ for GL261 and $P = 0.29$ for U87), a strong tendency towards improvement in the anti-VEGF regimen by anti-EGFL7 was observed. As a matter of fact, most vascular parameters measured displayed an additive effect upon combination therapy. The arrangement of the data in Fig. 6 and Suppl. Fig. 6 may have been misleading, so therefore we rearranged these figures for the sake of clarity. The combinatorial treatments are now shown as last data point in each line.

The additive effect of anti-EGFL7 on anti-VEGF treatment is further strengthened by our new combinatorial paradigm, where we applied both antibodies together with the chemotherapeutic agent temozolomide (new Fig. 6 K). EGFL7 significantly increased the median survival under this condition suggesting a beneficial effect of anti-EGFL7 for standard glioma therapy.

Furthermore, the EGFL7 protein holds some promising advantages over VEGF as a target molecule. EGFL7 is a secreted and barely soluble protein (comparable to fibronectin) which resides within the extracellular matrix of the tumor once the blood vessels are gone upon therapy. There it supports the regrowth of blood vessels along the trails of the former blood vessels during tumor recurrence. Thus, blocking EGFL7 slows blood vessel re-growth. In particular, this is relevant for malignant glioma as in this type of neoplasm usually the recurrence kills the patient, not the primary tumor. Therefore, anti-EGFL7 treatment does not only act comparably to anti-VEGF treatment (and likely improves it) but in addition offers some remarkable advantages over it.

In sum, anti-EGFL7 treatment might be used to reduce cytotoxicity of anti-VEGF in combinatorial regimens or it may even serve as an alternative treatment for anti-VEGF treatment itself. Therefore, we are convinced that our work is of major medical relevance for patients suffering from non-curable malignant glioma.

2) The mechanistic part, showing the effects of EGFL7 on integrin recycling and in vitro endothelial sprouting, is limited in scope. On one hand, the data provide only an incremental step in knowledge compared to previous results by the same group, and on the other they fail to address the actual relevance of these findings for glioma therapy. Does targeting of $\alpha 5\beta 1$ recapitulate the effects of anti-EGFL7 treatment in experimental glioma models? Or is the effect of EGFL7 targeting in gliomas due to other EGFL7 functions that have been described, for example its regulation of Notch signaling?

Response to the reviewer

The reviewer's opinion is well taken; however, understanding how integrin $\alpha V\beta 3$ affects intracellular trafficking of other integrins such as $\alpha 5\beta 1$ is neither trivial nor irrelevant. If we want to understand why treatment with promising drugs such as the $\alpha V\beta 3$ inhibitor cilengitide does not translate into a better prognosis or additional survival of glioma patients we need to understand how different integrins affect each other. This was in itself a challenging task and to understand in addition how EGFL7 influences this nexus is, in our view, a major step forward.

As described above in response to Reviewer 1, Point 3, in order to strengthen the understanding of the molecular mechanism behind our observations we analyzed Rho GTPase signaling downstream of integrins. These new results support our hypothesis on EGFL7's influence on coordinated

$\alpha V\beta 3/\alpha 5\beta 1$ trafficking (Fig. 4F-I). Furthermore, upon tumor engraftment of GL261 glioma cells ectopically expressing EGFL7, mice treated with an $\alpha 5\beta 1$ integrin inhibiting antibody exhibited significantly increased (by about 4.5 d) survival times (Fig. 5H). Clearly, these data show that EGFL7 affects glioma formation dependent on both integrins $\alpha V\beta 3$ and $\alpha 5\beta 1$.

3) The proposed molecular mechanism that EGFL7 increases surface expression of $\alpha 5\beta 1$ integrin through ligation with $\alpha V\beta 3$ makes sense, but it would need to be proven by a loss-of-function experiment, showing that interference with $\alpha V\beta 3$ by a blocking antibody or siRNA knockdown does prevent the effects of EGFL7 on $\alpha 5\beta 1$.

Response to the reviewer

As suggested, we have performed additional experiments in HUVECs using either the siRNA-based knock-down of integrin $\alpha V\beta 3$ or an $\alpha V\beta 3$ -specific blocking antibody. Interference with integrin $\alpha V\beta 3$ reduced the EGFL7-induced upregulation of surface integrin $\alpha 5\beta 1$ in primary endothelial cells. Unfortunately, the treatments themselves affected integrin $\alpha 5\beta 1$ in the absence of EGFL7, thus rendering these results inconclusive. At this stage, we see no possibility to circumvent this technical problem and therefore have not incorporated this data into the manuscript.

4) The data show that EGFL7 targeting reduces vessel maturation and increases permeability, but no attempt is made to address how EGFL7 may regulate these processes.

Response to the reviewer

In order to analyze the influence of EGFL7 on vessel maturation and permeability we performed several MRI studies to measure Gadovist extravasation and stained for the recruitment of mural cells and basal membrane deposition. Further, we provided a model on the interplay between $\alpha V\beta 3$ and $\alpha 5\beta 1$ integrin to molecularly describe our observations. We now include additional molecular data on how EGFL7 modulates GTPase signaling downstream of integrins in endothelial cells (Fig. 4F-I).

Previously, the physiological role of EGFL7 for the formation of the blood-brain barrier was studied by Schmidt et al. (Development, 2007). Recently, we investigated the role of EGFL7 in pathological blood-brain barrier formation in multiples sclerosis (Larochelle et al., Nat Commun, 2018). The reviewer is right that further studies on how EGFL7 affects vessel maturation and permeability would be very interesting, but the pathological environment of malignant glioma does not seem suitable for this purpose. Preferentially, this should be done in a physiological setting using different tools and mouse lines than the ones applied in this study and is therefore beyond the scope of this manuscript.

Other points

1) Fig. 1B - It is unclear how to interpret this panel: why do red circles appear to indicate methylation in CpG island 1 and no methylation in CpG island 2?

Response to the reviewer

The red circles demarcate tumor entities that differ from the gross of the tumor samples analyzed. CpG island 1 is found mostly unmethylated except in some samples of the RTKII subgroup (red circle labeled by one star). Conversely, CpG island 2 is found mostly methylated except in the G34R subgroup (red circle labeled by two stars). This is also described in Figure legend 1B: "Methylation arrays of primary glioblastoma (GBM) specimens revealed that CpG island 1 (egfl7 promoter) was mostly unmethylated with the exception of some samples in the RTKII subgroup (*). CpG island 2 (miR-126 promoter) was found methylated in most cases except the G34R subgroup (**)."

2) Suppl. Fig. 1D - It would be useful to include a scale also in the lower part of the y axis to interpret the range of the black bars: do they reach to 1? 2? 5?

Response to the reviewer

Suppl. Fig. 1D has been changed according to the reviewer's suggestion.

3) Suppl. Fig. 1E - I could not find any method for the derivation of organotypic spheroids from patient-derived GMB biopsies.

Response to the reviewer

An appropriate paragraph on patient-derived xenografts (PDX) has been added to the materials and methods section.

4) Suppl. Fig. 1F - The EGFL7 stains are difficult to interpret, possibly due to low magnification, and also appear to delineate dot-like structures rather than vessels, like evident instead in the histochemistries shown in Fig. 1E. Higher-magnification and clearer images would be helpful.

Response to the reviewer

The figure has been improved in order to address the reviewer's concern.

5) The conclusion that EGFL7 expression in glioma specimens was restricted to blood vessels is substantiated by the data, but that it occurred independently of miR-126/126 would be strengthened by similar analysis of its pattern of expression (or lack thereof) in the patient samples by in situ hybridization.*

Response to the reviewer

This is an excellent suggestion, however, the expression of EGFL7 and miR126 might be independent of each other but not necessarily exclusive. In order to address the reviewer's concern we applied two additional mouse models. First, we reduced EGFL7 expression in BTPC11 glioma cells using a shRNA-based knocked-down approach and implanted these cells into the striatum of immune-deficient mice. Second, we applied a novel EGFL7 knock-out mouse model, which we recently developed and which allows for the specific removal of EGFL7 from blood vessels in the absence of miR126 reduction (EGFL7^{fl/fl};Cdh5-CreERT2). These mice were intrastrially implanted with GL261 glioma cells. Both models revealed an increase in survival of tumor-bearing mice upon the reduction or loss of EGFL7 expression (new Figs. 2G+H), strengthening our point of EGFL7 acting as an oncogene and independent of miR126.

6) Suppl. Fig. 2B, C and F - The pattern of EGFL7 appears clearly suggestive of vascular structures, although the glioma cells are responsible for its overexpression in these over-expressing tumors. Curiously, instead, the pattern in panel 2E is not vascular anymore. This localization should be verified by co-staining with endothelial and glioma markers and an explanation would be useful.

Response to the reviewer

The structures staining positive for EGFL7 belong to the extracellular matrix surrounding the tumor cells. In the tumor bulk we did not observe a comparable massive amount of blood vessels, which are rather detected in the peritumoral rim of the glioma mass in our models. As a secreted protein, EGFL7 is transported via the endoplasmic reticulum and the golgi apparatus to the cellular exterior and deposited there in the extracellular matrix (Schmidt et al., Development, 2007), comparable to fibronectin. Neither GL261 nor U87 cells express endogenous EGFL7; therefore, these cells have been genetically engineered to ectopically express human or mouse EGFL7. Both proteins have been immunohistochemically stained post intracranial implantation as a proof of concept in order to validate stable EGFL7 expression during the course of the experiments. The detection of endogenous EGFL7 in the blood vessels of the mouse is notoriously difficult and works only with a few antibodies. Unfortunately, the antibodies used in this experiment were able to discriminate between recombinant human and mouse EGFL7 but did not pick up the signal of the endogenous vascular EGFL7, which remained invisible in this assay.

7) Fig. 3D-E - The provided MRI images appear to serve just a decorative function, since only one condition is shown and it is not actually mentioned which one it is. On the other hand, the graph in panel A is rather superfluous (intrastriatal injection and subsequent histological analyses are straightforward and were already described in a cartoon in Fig. 2A) and it could be removed, using the space to show complete MRI data for all conditions.

Response to the reviewer

According to the reviewer's suggestions, we included coronal MR images of all conditions (Fig. 3C+D), including T2-weighted MR images (delineation of the tumor in the right hemisphere) as well as T1-weighted images (assessment of contrast media leakage in tumors). These representative slices have been obtained from a stack of images covering the complete brain and from a larger cohort of data.

8) Figs. 2 and 5 analyze the same experiment and would best follow each other rather than being separated by the integrin data.

Response to the reviewer

The order has been switched according to the reviewer's suggestion.

9) *How were histological quantifications of vascular parameters actually performed? Vessel density is presented as cubic microns, which seems more a unit of volume than of density. What are the arbitrary units used to score vascular maturation? This information is missing from the methods and quite important to understand what is being measured and its significance. In fact, for example, just looking at the immunofluorescence images in Fig. 3 and Suppl. Fig. 3, pretty much all endothelial structures appear associated with collagen IV, SMA and PDGF-Rb signals in all conditions (except the single vessel shown in the control of Fig. 3G). This would suggest that there are no "naked vessels", but then quantifications in Arbitrary Units show dramatic differences for all these parameters.*

Response to the reviewer

Due to the heterogeneous vasculature in brain tumors we chose to analyze 3D image stacks, which yielded more precise quantifications as compared to 2D slices. Therefore, confocal microscopy images were acquired as stacks consisting of five confocal scans, which covered the depth of 250 μm of a histological section in total. The intensity of fluorescent staining of the endothelial cell marker CD31 was quantified by Imaris software and, due to the 3D reconstructions, calculated as the average of volume sums with the results plotted in cubic microns. Not every image may mirror the total result which emerged from a large amount of analyzed images. The pictures were intended to allow the reader to judge the staining quality and structures chosen for analysis during the experimental procedure. They have not been selected to represent the individual bars, which could only be judged on visualizing the complete 3D stack. However, in order to address the reviewer's concern images have been replaced by ones that more closely resembled the total result (Fig. 3 and Suppl. Fig 3). Vessel maturation was estimated according to the abundance of pericytes (PDGFR β), smooth muscle cells (SMA) and basement membrane (Col IV) accompanying blood vessels, which were identified by CD31 staining. Arbitrary units represent the fluorescence intensity sums of PDGFR β , SMA or Col IV in close proximity to CD31. We have now added this information to the material and methods section.

3rd Editorial Decision

19 June 2018

Thank you for the submission of your revised manuscript to EMBO Molecular Medicine. We have now received the enclosed report from the referee asked to re-assess it. As you will see the reviewer is now globally supportive and I am pleased to inform you that we will be able to accept your manuscript pending the following final amendments:

1) Please address the minor text change commented by referee 2.

Please address both referees' comments in writing. At this stage, we'd like you to discuss referee's 1 points and if you do have data at hand, we'd be happy for you to include it, however we will not ask you to provide any additional experiments at this stage.

Please provide a letter INCLUDING my comments and the reviewer's reports and your detailed responses to their comments (as Word file).

I look forward to reading a new revised version of your manuscript within 2 weeks.

***** Reviewer's comments *****

Referee #2 (Remarks for Author):

The Authors have performed several new experiments to address the weaknesses that had been identified. In particular, the new data, showing that combined anti-VEGF and anti-EGFL7 treatment significantly improves efficacy of a standard chemotherapeutic regimen, overcomes the previous

limitations in medical impact. The new experiments on the effects of integrin manipulation in vivo and their role in the mechanism of action also nicely extend and complete the extent of the findings.

Just a couple of very minor clarifications might be useful for the readers:

- The Authors mention the selection of new immunofluorescence images depicting vessel maturation in Figure 3, to better reflect the quantified values in the graphs below. However, it appears to me that the new Figure file still contains the same images as the first version.

- In the new panel 6K, it would be useful to explicitly clarify, both in the legend and the results text, that chemotherapy is given in all conditions as a baseline, and that therefore "control" actually means chemotherapy alone.

3rd Revision - authors' response

02 July 2018

Editor (Remarks for Author):

1) Please address the minor text change commented by referee 2. Please address both referees' comments in writing. At this stage, we'd like you to discuss referee's 1 points and if you do have data at hand, we'd be happy for you to include it, however we will not ask you to provide any additional experiments at this stage. Please provide a letter INCLUDING my comments and the reviewer's reports and your detailed responses to their comments (as Word file).

Response to the editor

Please find the discussion of the reviewer's points below.

Referee #2 (Remarks for Author):

The Authors have performed several new experiments to address the weaknesses that had been identified. In particular, the new data, showing that combined anti-VEGF and anti-EGFL7 treatment significantly improves efficacy of a standard chemotherapeutic regimen, overcomes the previous limitations in medical impact. The new experiments on the effects of integrin manipulation in vivo and their role in the mechanism of action also nicely extend and complete the extent of the findings.

Just a couple of very minor clarifications might be useful for the readers: The Authors mention the selection of new immunofluorescence images depicting vessel maturation in Figure 3, to better reflect the quantified values in the graphs below. However, it appears to me that the new Figure file still contains the same images as the first version.

Response to the reviewer

These changes affected the control panels in Fig. 3G+I. However, once again we'd like to point out that these are just representative pictures to allow for judging on staining quality and structures chosen for subsequent analysis. Quantifications illustrated in the corresponding graphs have been made by usage of 3D stacks.

In the new panel 6K, it would be useful to explicitly clarify, both in the legend and the results text, that chemotherapy is given in all conditions as a baseline, and that therefore "control" actually means chemotherapy alone.

Response to the reviewer

The term "control" in Figure 6 has been substituted by "TMD", the abbreviation for temozolomide, as suggested by the reviewer.

Corresponding Author Name: Mirko HH Schmidt

Manuscript Number: EMM-2017-08420-V3